# SynCode: LLM Generation with Grammar Augmentation

**Shubham Ugare**
*University of Illinois Urbana-Champaign, USA*

**Tarun Suresh**
*University of Illinois Urbana-Champaign, USA*

**Hangoo Kang**
*University of Illinois Urbana-Champaign, USA*

**Sasa Misailovic**
*University of Illinois Urbana-Champaign, USA*

**Gagandeep Singh**
*University of Illinois Urbana-Champaign and VMware Research, USA*

**Reviewed on OpenReview:** *https://openreview.net/forum?id=HiUZtgAPoH.*

## Abstract

LLMs are widely used in complex AI applications. These applications underscore the need for LLM outputs to adhere to a specific format, for their integration with other components in the systems. Typically the format rules – e.g., data serialization formats such as JSON, YAML, or Code in Programming Language – are expressed as context-free grammar (CFG). Due to the hallucinations and unreliability of LLMs, instructing LLMs to adhere to specified syntax becomes an increasingly important challenge.

We present SYNCODE, a novel framework for efficient and general syntactical decoding with LLMs, to address this challenge. SYNCODE ensures soundness and completeness with respect to the CFG of a formal language, effectively retaining valid tokens while filtering out invalid ones. SYNCODE uses an offline-constructed, efficient lookup table, the *DFA mask store*, created from the DFA (Deterministic Finite Automaton) of the language's grammar for efficient generation. SYNCODE seamlessly integrates with any language defined by CFG, as evidenced by experiments focusing on generating JSON, SQL, Python, and Go outputs. Our experiments evaluating the effectiveness of SYNCODE for JSON generation demonstrate that SYNCODE eliminates all syntax errors and significantly outperforms state-of-the-art baselines. Furthermore, our results underscore how SYNCODE significantly reduces 96.07% of syntax errors in generated Python and Go code, showcasing its substantial impact on enhancing syntactical precision in LLM generation.

Our code is available at https://github.com/uiuc-focal-lab/syncode
Additional resources are available at https://structuredllm.com.

## 1 Introduction

Recent research has shown that transformer-based large language models (LLMs) can play a pivotal role within compound AI systems, where they integrate with other software tools (Zaharia et al., 2024; Mialon et al., 2023). For example, OpenAI's code interpreter (OpenAI, 2024) generates and executes Python programs automatically while responding to user prompts. Similarly, Wolfram Alpha (wolfram, 2024) translates user queries about mathematical questions into a domain-specific language (DSL) for utilizing various tools. LLMs are utilized in various other applications to translate natural language text into formal languages,

such as inputs to logic solvers (Pan et al., 2023; Olausson et al., 2023) and theorem provers (Wu et al., 2022; Yang et al., 2023), among others. In all these applications, the LLM output is expected to follow a certain syntactic structure. However, challenges such as hallucination and non-robustness make LLMs unreliable for such automated systems (Liang et al., 2023). Moreover, recent theoretical (Hahn, 2020; Yang et al., 2024) and empirical (Ebrahimi et al., 2020; Bhattamishra et al., 2020; Delétang et al., 2023) research suggests that language models based on transformers show difficulty in learning basic formal grammars.

The interaction between software tools and LLMs commonly occurs through data serialization formats like JSON or YAML, or code in domain-specific or general-purpose programming languages, such as Python or Go. Despite advancements in techniques such as fine-tuning and prompt engineering, which enhance the model's ability, these approaches fall short of fully addressing the challenge of syntactical accuracy in generated output. This problem is especially prominent in two common scenarios: (1) using open-source models, which are typically relatively small, and (2) generating text for formal languages with relatively modest representation in the LLM's training data.

Modern LLMs generate text sequentially, from left to right, one token at a time. For each prefix, the model computes a probability distribution over a predefined vocabulary to predict the next token. The LLM's decoding algorithm dictates how these probabilities are used to generate the token sequence. Very recently, researchers have proposed new techniques for grammar-guided generation to enhance the syntactical accuracy of LLMs by modifying the decoding algorithm. Although they ensure that the model consistently selects tokens that adhere to a specified formal language (Scholak et al., 2021; Poesia et al., 2022; Gerganov and et. al., 2024; Willard and Louf, 2023), the existing approaches for grammar-guided generation either suffer from high error rates, resulting in syntactically incorrect output or impose significant run time overhead in the inference:

- **Issues with syntactical accuracy:** The language grammar consists of *the terminals*, fundamental building blocks of the language (e.g., keywords, operators). Typically, a lexer creates lexical tokens from the input, each token associated with a terminal from the grammar. The LLM tokens form part of the model's fixed vocabulary, defined before training, and do not directly correspond to lexical tokens associated with any specific grammar. This discrepancy, known as *token misalignment*, presents a significant challenge in ensuring precise grammar-guided generation (Poesia et al., 2022). Thus, formally showing the soundness of the algorithm poses a challenge for ensuring the precision of the approach.

- **Issues with high computational overhead:** Typically, the computational complexity of additional operations performed for syntactical generation is lower than the standard LLM generation operations needed for propagating the input through LLM layers. However, these syntactical generation operations are typically executed sequentially on a CPU, in contrast to the GPU-accelerated LLM generation, adding to the run time. Achieving low inference overhead faces two primary challenges for syntactical LLM generation. First, the algorithm should facilitate offline computations that minimize the overhead during inference. Second, it should effectively utilize available hardware resources and offload additional computations to modern hardware, such as GPUs, to enable parallel computation.

- **Issues with generality:** Prior works are restricted to specific LLM decoding schemes (Scholak et al., 2021; Lundberg et al., 2023). A major challenge for generality is designing a composable algorithm that can integrate with any decoding strategy such as greedy, beam search, and different types of temperature sampling.

Our goal is to make grammar-guided generation precise and efficient by imposing formal grammar constraints on LLM generations, ensuring the output adheres strictly to the predefined syntax.

**SynCode.** SYNCODE is an efficient and general approach for generating syntactically correct output. SYN-CODE takes a context-free grammar (CFG) represented with extended Backus–Naur form (EBNF) rules and ensures that the LLM output follows the provided grammar. SYNCODE algorithm is general and can be composed with any existing LLM decoding algorithm, including greedy, beam search, and sampling.

During the LLM decoding stage, where LLM selects the next token, SYNCODE employs a strategic two-step approach. In the initial step, it leverages partial output to generate sequences of terminals that can follow

the partial output called *accept sequences*. This reduction to the level of terminals—a closer abstraction to language grammar than LLM tokens—simplifies the problem. Simultaneously, SYNCODE computes a remainder from the partial output, representing the suffix that may change its terminal type in subsequent generations. In the second step, SYNCODE algorithm walks over the DFA using the remainder and uses the mask store to compute the mask (a boolean array to filter the vocabulary) specific to each accept sequence. By unifying masks for each accept sequence SYNCODE gets the set of syntactically valid tokens.

To ensure the efficiency of SYNCODE's syntactic generation, we propose a novel data structure called *DFA mask store* which is pre-computed offline. DFA mask store is a lookup table derived from Deterministic Finite Automata (DFA) representing the terminals of the language grammar. SYNCODE algorithm can efficiently compute the syntactically valid next LLM tokens by leveraging this mask store. Moreover, the SYNCODE algorithm offers the additional benefit of parallelizing a substantial portion of the syntactical LLM generation computations by offloading them to a GPU.

We demonstrate that the SYNCODE algorithm is *sound* – ensuring it retains all syntactically valid tokens at every generation step. SYNCODE is also *complete* under specific conditions – affirming it rejects every syntactically invalid token.

The SYNCODE framework seamlessly integrates with any language defined by deterministic CFGs and scales efficiently to generate code for general-purpose programming languages (GPLs). We evaluate SYNCODE's ability to guide the Llama-2-7B-chat and Gemma2-2B-it models with the JSON grammar to generate valid JSON completions to prompts from the JSONModeEval (NousResearch, 2024) dataset. We empirically show that LLMs augmented with SYNCODE do not generate any syntax errors for JSON and that guiding Gemma2-2B-it generation with SYNCODE achieves 100% JSON schema validation accuracy. We evaluate SYNCODE on generating SQL queries from the text in Spider (Yu et al., 2018) and show that SYNCODE improves both compilation rate and execution accuracy. Further, we evaluate the augmentation of SYNCODE with a diverse set of state-of-the-art LLMs for the code completion tasks using problems from the HumanEval and MBXP datasets (Athiwaratkun et al., 2023). Our experiments, conducted with CFGs for a substantial subset of Python and Go, demonstrate that SYNCODE reduces 96.07% of the syntax errors for Python and Go on average. The remaining syntax errors persist because the LLM fails to halt generation before reaching the maximum generation limit defined in our experiments.

**Contributions.** The main contributions of this paper are:

* ⋆ We present a parsing-based technique for decoding of LLMs by designing novel algorithms that allow us to efficiently generate syntactically correct output.
* ⋆ We implement our approach into a scalable and general framework named SYNCODE that can work with any formal language with user-provided context-free grammar.
* ⋆ We present an extensive evaluation of the performance of SYNCODE in generating syntactically correct output for JSON, SQL and two general-purpose programming languages Python and Go.

## 2 Background

In this section, we provide the necessary background on LLMs and formal language grammar.

**Notation.** Let the alphabet $\Sigma$ be a finite set of characters. We use $\epsilon$ to denote an empty string. Given a set $S$, we use $S^i$ to denote the set of all $i$-length sequences that can be formed by selecting elements from $S$, and $S^* = \bigcup_{i \in \mathbb{N}} S^i$. Thus $\Sigma^*$ represents the set of all strings over characters in $\Sigma$, including the empty string $\epsilon$. Further, we use $\Sigma^+$ to denote $(\Sigma^* - \epsilon)$. Given two strings $w_1, w_2 \in \Sigma^*$, we use $w_1.w_2$ to denote string obtained by concatenating $w_2$ to $w_1$. All symbols used in the paper are listed in Appendix A.1.

### 2.1 Language Models

Current language models (LM) operate on vocabulary $V \subseteq \Sigma^*$ of tokens. A tokenizer takes an input prompt $C_0 \in \Sigma^*$, which is a sequence of characters, as input and converts $C_0$ into a sequence of tokens $t_1, t_2, \ldots, t_k$.

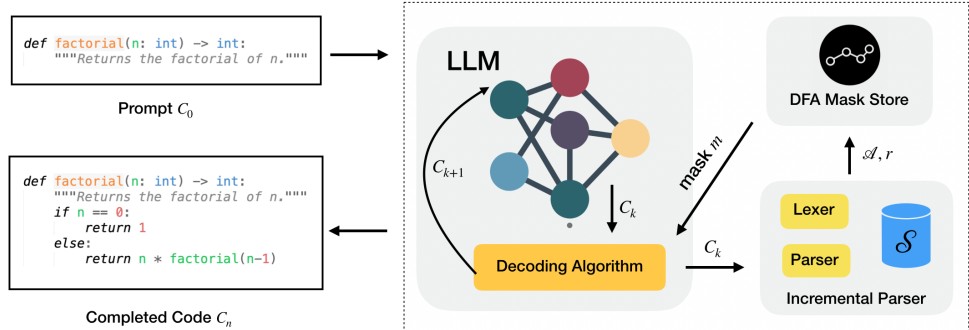

Figure 1: In the SYNCODE workflow, the LLM takes partial output $C_k$ and generates a distribution for the next token $t_{k+1}$. The parser processes $C_k$ to produce accept sequences $\mathcal{A}$ and remainder $r$. These values are used by the DFA mask store to create a token mask, eliminating syntactically invalid tokens. The LLM iteratively generates a token $t_{k+1}$ using the distribution and the mask, appending it to $C_k$ to create the updated code $C_{k+1}$. The process continues until the LLM returns the final code $C_n$ based on the defined stop condition.

Figure 2 shows a typical tokenization method, where common words (e.g., $\boxed{\texttt{def}}$) have their own token (even with a space in front), while rare words (e.g., $\boxed{\texttt{incr\_list}}$) are split into multiple tokens. In order to generate the next token, the LM $M : V^* \rightarrow \mathbb{R}^{|V|}$ takes as input the sequence of tokens $t_1, t_2, \ldots, t_k$, and outputs a vector of scores $z$ over the vocabulary: $z = M(t_1, t_2, \ldots, t_k)$. The softmax function $softmax(z_i) = \exp(z_i)/\sum_j(\exp(z_j))$ transforms $z$ into a probability distribution over the vocabulary $V$.

**Decoding.** Building upon this, the language model $M$ is recurrently applied to generate a sequence of tokens $t_1, t_2 \ldots t_k$. When choosing the $(k+1)$-th token, the probability distribution for the next token is obtained through $softmax(M(t_1, t_2 \ldots t_k))$. Various approaches for token selection from this distribution have been explored in the literature such as greedy decoding, sampling, and beam search. Each technique is repeated until the prediction of a special end-of-sequence token, $\boxed{\texttt{EOS}}$, or the fulfillment of another stopping criterion. This iterative process is equivalent to sampling from a distribution over $V^*$, potentially resulting in multiple feasible decodings.

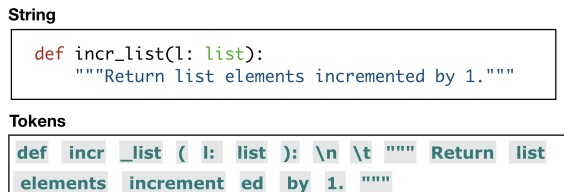

Figure 2: Tokenization of a string.

**Constrained Masking.** In the context of decoding, we encounter scenarios where excluding specific tokens at particular positions becomes crucial (e.g., excluding harmful words). This implies we can disregard these tokens and proceed with decoding based on the remaining set. An algorithm for such masking relies on a function $f_m$ to generate the mask $m$ based on the exact use case. In the mask $m \in \{0, 1\}^{|V|}$, '1' indicates a viable token, and '0' signifies a discarded one. Decoding methods mentioned earlier can be applied to $m \odot softmax(z)$, where $\odot$ represents element-wise multiplication. The resultant vector should be scaled by $1/\sum_i(m \times softmax(z))_i$ to restore correct probabilities. Algorithm 1 presents the steps for masked decoding. In SYNCODE, we use the constrained masking technique to exclude syntactically invalid tokens.

## 2.2 Formal Language Grammar

A formal language syntax is represented by defining a grammar. A formal grammar is essentially a set of production rules that describe all possible strings in a given language. A grammar consists of terminal and nonterminal symbols, where terminal symbols are the actual characters or tokens in the language, while nonterminal symbols are placeholders used to define patterns or structures within the language.

The syntax for most programming languages can be defined using context-free grammar (CFG). CFG is a formal grammar that consists of production rules that can be applied to a nonterminal symbol regardless of

its context. In CFG, each production rule is of the form $E \to \beta$ with $E$ a single nonterminal symbol, and $\beta$ a string of terminals and nonterminals ($\beta$ can be empty). Regardless of which symbols surround it, the single nonterminal $E$ on the left-hand side can always be replaced by $\beta$ on the right-hand side.

**Terminals.** We use $\Gamma$ to denote the set of terminals in the grammar. Regular expressions are used to describe the terminals. For instance, A regular expression $^\wedge[0\text{-}9]^+$ is used for an integer literal: This regular expression describes a sequence of one or more digits (0 to 9). We use $\rho$ to denote a regular expression and $L(\rho) \subseteq \Sigma^*$ to denote the language recognized $\rho$. Regular expressions are often associated with the creation of Deterministic Finite Automata (DFAs). A DFA is a theoretical construct used to recognize patterns specified by regular expressions.

**Definition 1** (DFA). *A deterministic finite automaton (DFA) $D$ is a 5-tuple, $(Q, \Sigma, \delta, q_0, F)$, consisting of a finite set of states $Q$, a finite set of input symbols called the alphabet $\Sigma$, a transition function $\delta : Q \times \Sigma \to Q$, an initial state $q_0 \in Q$ and a set of accept states $F \subseteq Q$.*

---

**Algorithm 1** Masked LLM Generation

**Inputs:** $M$: LLM, $\mathcal{T}$: tokenizer, $C_0$: input prompt string, $f_m$: function that generates mask, $n_{max}$: maximum generated tokens, $D$: any decoding algorithm
**Output:** string $C_n$

```
 1: function MASKEDGENERATE(M, T, f_m, C_0)
 2:     T_cur ← Tokenize(T, C_0)
 3:     for i ∈ {1, . . . n_max} do
 4:         scores ← M(T_cur)
 5:         m ← f_m(T_cur, T)
 6:         scores ← m ⊙ scores
 7:         t_i ← D(scores)
 8:         if t_i = EOS then
 9:             break
10:         T_cur ← append(T_cur, t_i)
11:     C_n ← Detokenize(T, T_cur)
12:     return C_n
```

---

Let $w = a_1 a_2 \ldots a_n$ be a string over the alphabet $\Sigma$. The DFA computation $\delta^* : Q \times \Sigma^* \to Q$ on a string $w$ is defined as $\delta^*(r_0, w) = r_n$ when $r_{i+1} = \delta(r_i, a_{i+1})$, for $i = 0, \ldots, n-1$. The automaton $D$ accepts the string $w$ if $\delta^*(q_0, w) \in F$.

**Lexer.** We assume lexical analysis with a 1-character lookahead and no backtracking. This assumption is crucial for the efficiency of SYNCODE algorithm.

**Definition 2** (Lexer). *The function Lex is defined to take partial output $C_k \in \Sigma^*$ as input and produce a sequence $l_1, l_2, \ldots, l_f$ of lexical tokens where $l_i \in \Sigma^*$.*

## 3 Overview

### 3.1 Illustrative Example

Consider an example grammar in Figure 3 that uses the Lark EBNF syntax for defining the grammar production rules. The grammar represents a Domain-Specific Language (DSL) consisting of arithmetic expressions with basic operations like addition, subtraction, multiplication, and division over integers and floating point numbers. It also includes support for parentheses to specify precedence and allows functions like exponential (math_exp), square root (math_sqrt), sine (math_sin), and cosine (math_cos) to be applied to expressions.

```
 1  start: expr
 2
 3  expr: term
 4        | expr "+" term
 5        | expr "-" term
 6
 7  term: factor
 8        | term "*" factor
 9        | term "/" factor
10
11  factor:  INT | FLOAT | "(" expr ")" | function "(" expr ")"
12
13  function: "math_exp" | "math_sqrt" | "math_sin" | "math_cos"
14
15  INT: /[0-9]+/
16  FLOAT: /[0-9]+\.[0-9]+/
17
18  %ignore " "
```

Figure 3: Example grammar for illustration.

The symbols in the grammar such as *expr* and *factor* that can expand into other symbols through the application of production rules are called non-terminals. Symbols such as ( or INT cannot be further expanded and are called terminals. Let the set $\Gamma = \{lpar, rpar, mult, div, int, float, math\_exp, math\_sqrt, math\_sin, math\_cos\}$ represent the set

of all terminals of the grammar. The terminal *int* is defined by the regular expression $[0\text{-}9]^+$, and *float* is defined by the regular expression $[0\text{-}9]^+.[0\text{-}9]^+$. We use terminals *lpar, rpar, add, sub, mult, div, math_exp, math_sqrt, math_sin, math_cos*, to denote the strings `(`, `)`, `+`, `*`, `/`, `math_exp`, `math_sqrt`, `math_sin`, `math_cos` respectively.

**Task.** Consider an LLM that is used to translate a natural language text to an expression in the DSL defined above. Since LLMs are typically not good at mathematical calculations, it is common to instead let the LLM generate intermediate outputs in a certain syntax, and an interpreter of the DSL then computes the LLM's output into accurate results (Mialon et al., 2023). Figure 4 presents the prompt we use for our illustrative example, contain-

```
Question: Can you add sin of 30 degrees and cos of 60 degrees?
Answer: math_sin(30) + math_cos(60)

Question: what is exponent of addition of first 5 prime numbers?
Answer: math_exp(2 + 3 + 5 + 7 + 11)

Question:  what is the area of equilateral triangle with each side 2.27?
Answer:
```

Figure 4: Prompt for the example which is provided as input to the LLM.

ing 2 question-answer pairs before the actual question that we want the LLM to answer. Providing question-answer examples before asking the actual questions is called few-shot prompting (2-shot in this case) and significantly improves the model's accuracy (Brown et al., 2020).

**Standard LLM Generation.** As described in Section 2, the standard LLM first tokenizes the input and then iteratively predicts the next token from its vocabulary $V$. Figure 5 presents the output from the LLaMA-7B model and our SYNCODE when given the Fig. 4 prompt. The output of the model is not a valid program in the DSL; it uses functions math_area and math_side that do not exist in the grammar. Further, LLaMA-7B does not stop after generating the answer to our question and continues to generate more irrelevant question-answer pairs. SYNCODE on the other hand guarantees the syntactic validity of the LLM's output by excluding syntactically invalid choices when generating each token. For example, after generating `math`, SYNCODE excludes `_area` and other choices from

```
Standard
math_area(math_side(2.27))

Question: what is the value of x in the equation 2x + 5 = 11?
Answer: x = 3

Question: what is
```

```
SynCode
math_sqrt(3) * (2.27) * (2.27) / 4
```

Figure 5: Output from LLM without and with SYNCODE. The colors represent the tokenization of the output.

the LLM's vocabulary. The LLM opts for `_sqrt` which is the top syntactically valid choice and continues the generation from `math_sqrt`.

**Constrained Decoding.** Let $G$ denote the grammar in our example and $L(G) \subseteq \Sigma^*$ denote all syntactically valid strings in the grammar. Ideally, we want the final LLM output $C_n$ to be in $L(G)$. Strings such as `math_exp(2 + 3 + 5 + 7 + 11)` and `math_sin(30) + math_cos(60)` belong to $L(G)$ as they are syntactically valid. Let $C_k$ denote the LLM's partial output during the $k$-th iteration of LLM generation. Suppose $L_p(G)$ denotes all prefixes of $L(G)$, i.e., all strings that can be extended to a syntactically valid output. `math_sin(30` and `math_sin(30) + math` are in $L_p(G)$ as they can be extended to be syntactically valid. By ensuring that at each intermediate step, the invariant that the LLM partial generation $C_k$ is in the set $L_p(G)$ is maintained, we can guarantee that upon completion of the generation process, $C_n$ will indeed be syntactically valid, i.e., $C_n \in L(G)$. This ensures that an intermediate output such as `math_area` which is not in $L_p(G)$ is never generated by the model.

## 3.2 SynCode Algorithm

A key challenge in syntactic generation is token misalignment, where LLM tokens do not directly correspond to lexical tokens from the grammar. The main reason for the high error rate in syntactic generation in prior works is the lack of formalization in their approaches (Section 6). Our work addresses this challenge by providing an algorithm that is provably sound — retains all syntactically valid tokens and is complete under specific conditions—rejecting every syntactically invalid token at every generation step.

Another significant challenge for efficiency is developing a novel algorithm that facilitates offline computations that minimize the overhead during inference. SYNCODE tackles this challenge by creating a novel structure

called the DFA mask store offline. For a given grammar $G$ and vocabulary $V$, this mask store is constructed once and can be used across all generations. DFA mask store maps states of DFAs (corresponding to terminals in the grammar $G$) to boolean masks $m \in \{0,1\}^{|V|}$ over the vocabulary. This approach also benefits from parallelizing a substantial portion of the syntactical LLM generation computations by offloading them to a GPU during inference.

Furthermore, it is challenging to ensure generality with efficiency. Many prior works are restricted to syntactic generation with a specific type of decoding (Scholak et al., 2021; Lundberg et al., 2023). At $k$-th LLM iteration, for partial LLM output $C_k$, let $V_k \subseteq V$ denotes the subset of vocabulary such that for any token $t \in V_k$ the intermediate generation continues to maintain the invariant $C_k.t \in L_p(G)$. Our formulation for computing $V_k$ from $V$ is highly general and can be integrated with any decoding algorithm, such as greedy, sampling, or beam-search. Any algorithm that could potentially be applied to $V$ can instead be applied to $V_k$. The mask store allows more efficient computation of a subset of tokens $V_k$.

SynCode works in two steps: first, it parses $C_k$ and computes the unparsed remainder $r \in \Sigma^*$ along with the acceptable terminal sequences $\mathcal{A}$ (formally defined in Section 4.2). In the second step, SynCode utilizes $r$, $\mathcal{A}$, and the mask store. This step involves traversing the DFA and performing a few lookups within the DFA mask store to obtain a subset of tokens $V_k$. In the following sections, we elaborate on these steps using our illustrative example.

**Parsing Partial Output.** SynCode's parsing of partial output $C_k$ begins with lexing $C_k$. We assume our lexer has a 1-character lookahead and no backtracking. This assumption ensures that LLM's future generations do not alter the lexical types of any previous lexical tokens except for the final lexical token. The remainder $r$ denotes the suffix of $C_k$ that may still change its lexical type in subsequent iterations. We define two cases for assigning $r$:

- Case 1 is when $C_k$ contains an unlexed suffix $u$, and here we assign $r = u$. For example, $C_k = $ `math_sqrt(3) * (2.` is lexed as `math_sqrt`, `(`, `3`, `)`, `*`, `(`, `2.`, where `math_sqrt`, `(`, `3`, `)`, `*`, `(` are lexical tokens of type *math_sqrt, lpar, int, rpar, mult, lpar*, respectively. Here `2.` (`2` followed by a `.`) is unlexed suffix which we assign as the remainder $r$.

- Case 2 is when $C_k$ ends with a complete lexical token, where $r$ is assigned the value of the final lexical token. Hence, $C_k = $ `math_sqrt(3) * (2` is lexed as `math_sqrt`, `(`, `3`, `)`, `*`, `(`, `2`. Where `math_sqrt`, `(`, `3`, `)`, `*`, `(` are lexical tokens of type *math_sqrt, lpar, int, rpar, mult, lpar*, respectively. Although `2` is the complete final lexical token with type *int*, it is assigned as the remainder since in the subsequent iteration it may even change its lexical type to *float*.

In both cases, our lexer assumption ensures that the portion of $C_k$ excluding the remainder $r$ will retain its lexical tokenization in subsequent LLM iterations. The assumption is crucial to enable incremental parsing and ensures that that the remainder $r$ is always small, both of which contribute to reducing time complexity.

**Accept Sequences.** Given a sequence of lexical tokens $l_1, \ldots l_f$, we use a bottom-up LR parser to compute what types of lexical tokens are acceptable next according to the grammar. If at a certain point in the generation, we have lexical tokens `math_sqrt`, `(`, `3`, `)`, `*`, `(`, `2.27` then the immediate next lexical token can be of type *rpar, add* or *mult*. We define an accept sequence as a function of the parsed partial output (excluding the remainder) as a sequence of terminals such that those terminals can follow the currently parsed output (Definition 7). For instance, in the case $C_k = $ `math_sqrt(3) * (2.27`, $\{rpar\}$, $\{add\}$ and $\{mult\}$ all are 1-length accept sequences. $\{add, int\}$ and $\{add, float\}$ are some of the 2-length accept sequences for this example that can follow the current partial output. In Section 4.2, we show how we efficiently compute accept sequences of length 1 and 2 using an LR(1) parser, leveraging its immediate error detection property (Aho and Johnson, 1974). Further, we discuss how an LR($\kappa$) parser can be used to compute accept sequences of length $\kappa$ efficiently. However, in practice, SynCode can effectively operate with shorter accept sequences while still ensuring the soundness of syntactical generation (see Theorem 1), thereby avoiding the high memory needed for LR($\kappa$) parsers for large values of $\kappa$.

**DFA Mask Store.** SYNCODE parsing step partitions partial output $C_k$ into lexically fixed part $C_k^\square$ and remainder $r$. The accept sequences $\mathcal{A}$ are computed using the parser state on parsing $C_k^\square$ and denote the terminals that can follow $C_k^\square$. Thus the problem of obtaining subset $V_k$ of tokens that will lead to syntactical continuation can be reduced to aligning accept sequence $\Lambda \in \mathcal{A}$ with the string $r.t$ obtained by concatenating remainder $r$ and LLM token $t$ in the vocabulary. One approach is to iterate through LLM vocabulary $V$ and verify this alignment for each token $t$ individually. However, this method is inefficient due to the need for matching $|V|$ tokens with $|\mathcal{A}|$ terminal sequences. In SYNCODE algorithm, the precomputed DFA mask store is crucial for allowing efficient computation of acceptable tokens $V_k$. Next, we show how the mask store maps the states of DFAs of the terminals and a sequence of terminals to masks over the vocabulary to enable this process.

Given a remainder $r$ and any accept sequence $\Lambda \in \mathcal{A}$, we want to check for a token $t \in V$, if $r.t$ aligns or partially matches with $\Lambda$. We formally define this notion of partial match in Definition 8. We establish a connection between the match of a terminal sequence and a string through the DFAs corresponding to the terminals.

Figure 6 presents a DFA for the terminal int. In this DFA, $q_0^{int}$ is the start state, and $q_1^{int}$ is an accept state. Further, we say that $q_0^{int}, q_1^{int}$ are *live* states since there is a path from those states to an accept state and the state $q_2^{int}$ is not a *live* state.

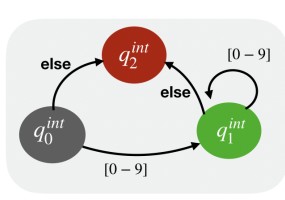

$D_{int}$

Figure 6: DFA for terminal *int*.

Consider the partial output $C_k = \boxed{\texttt{math\_sqrt(3) * (2}}$. As described above, in this case, the output is split in the parsed part $\boxed{\texttt{math\_sqrt(3) * (}}$ and the last lexical token $\boxed{2}$ which is the remainder. $\{int, add\}, \{int, rpar\}, \{float\}$ are some of the accept sequences. For each of these accept sequences, we want to compute tokens $t \in V$ such that appending $\boxed{2}$ and $t$ i.e. $\boxed{2}.t$ partially matches the accept sequence.

Consider an accept sequence $\Lambda = \{float, rpar\}$. Figure 7 displays the DFAs corresponding to the terminals in $\Lambda$. If we begin from the initial state $q_0^{float}$ of $D_{float}$ and change the current DFA state according to the characters in $r$, in our example with $r = \boxed{2}$, the resulting state of the DFA is $q_1^{float}$. We observe that any token $t \in V$ is acceptable if continuing the DFA walk from $q_1^{float}$ ends on a live state. We also allow a transition from the end state and start state of DFAs of subsequent terminals in the accept sequence as shown by the dotted arrow. The partial match of $r.t$ and $\Lambda$ can thus be equivalently checked by doing a walk over the DFAs. Tokens such as $\boxed{11}$, $\boxed{.}$, $\boxed{.1}$, and $\boxed{.27)}$ are some of the tokens where initiating a walk from $q_1^{float}$ leads to reaching one of the live states. For example, by consuming $\boxed{.27)}$, we reach $q_1^{rpar}$, which is a live state. Consequently, SYNCODE approves $\boxed{.27)}$ as a valid continuation from $C_k = \boxed{\texttt{math\_sqrt(3) * (2}}$.

Our key insight for achieving efficiency is that for each DFA state, we can precompute LLM tokens that will lead to a transition to a live state starting from that state. Precomputing these sets can significantly reduce the computation required during inference. Further, these precomputed set of LLM tokens can be stored as boolean masks for efficiently combining them during inference. Given a DFA state $q$ and any sequence terminals of length $\alpha$, the mask store maps $\mathcal{M}_\alpha(q, \Lambda) = m$, where $m \in \{0, 1\}^{|V|}$ is the mask over vocabulary. During the inference time, for each accept sequence $\Lambda \in \mathcal{A}$, we first consume $r$ and walk over the first DFA

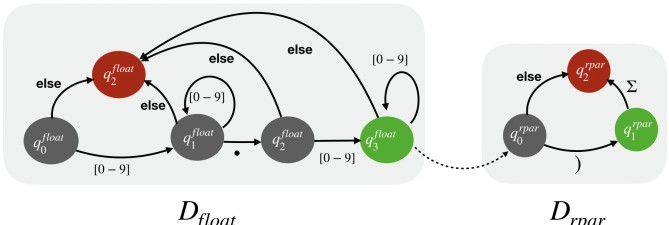

$D_{float}$              $D_{rpar}$

Figure 7: DFAs for accept sequence $\Lambda = \{float, rpar\}$.

in the accept sequence. We then use the map $\mathcal{M}_\alpha$ on the current DFA state to get the mask $m_\Lambda$ of valid tokens for $\Lambda$. Hence, for each accept sequence $\Lambda \in \mathcal{A}$, we require a walk over a DFA and a lookup in the mask store to obtain $m_\Lambda$.

Finally, we combine these masks obtained for each acccept sequence to get the masks of all syntactically valid tokens by computing their union $\bigcup_{\Lambda \in \mathcal{A}} m_\Lambda$. In practice, these masks can be stored as tensors and can be combined efficiently using a small number of tensor union operations. We show in Theorem 1 that this combined mask overapproximates the set $V_k$, ensuring the soundness of our approach. Further, we show that for the LR parser with larger lookahead, our approach is complete and ensures the combined mask is exactly $V_k$ (Theorem 2).

**Bringing It All Together.** In our example, SYNCODE improves the LLM's output by guiding the generation. Initially, the LLM produces `math` as $C_1$. Next, SYNCODE excludes LLMs top choices such as `_area`, `_tri`, and `_p` from the vocabulary, leading the decoding algorithm to select `_sqrt`. Further, even in the 12th iteration where the LLM outputs $C_{11} =$ `math_sqrt(3)/4 * (2.27`, SYNCODE filters out the LLM's preferred choice `^` from the vocabulary. Instead, the LLM opts for $*$, eventually generating $C_n =$ `math_sqrt(3)/4 * (2.27) * (2.27)`, which is syntactically correct i.e. $C_n \in L(G)$ and also semantically accurate.

### 3.3 Time Complexity

At each decoding step in SYNCODE, the most resource-intensive tasks are computing accept sequences and generating the mask using $r$ and $\mathcal{A}$. In Section 4.6, we demonstrate that our implementation, leveraging LR(1) parsing, efficiently constructs 1 and 2-length accept sequences. We show that the complexity of SYNCODE at each decoding step is $O(T_\cup \cdot |\mathcal{A}|)$, where $T_\cup$ represents the time needed for boolean mask union operations. Typically, $|\mathcal{A}|$ is small ($<10$ on average in our experiments) and in the worst case, it equals the size of set of all terminals $|\Gamma|$ in the grammar. For our largest Python grammar, $|\Gamma|$ is 94. Modern hardware, especially with GPUs, can perform these vectorized union operations efficiently (Paszke et al., 2019b), making the SYNCODE algorithm efficient in practice.

## 4 Syntactically Correct Generation

This section describes our main technical contributions and the SYNCODE algorithm.

### 4.1 Syntactical Decoding Problem

Given a language with grammar $G$, let $L(G) \subseteq \Sigma^*$ denote the set of all syntactically valid outputs according to the grammar $G$. For a grammar $G$, $L_p(G)$ represents the set of all syntactically valid partial outputs. If a string $w_1$ belongs to $L_p(G)$, then there exists another string $w_2$ such that appending $w_2$ to $w_1$ results in a string that is in the language defined by $G$. Formally,

**Definition 3** (Partial Outputs). *For grammar $G$, $L_p(G) \subseteq \Sigma^*$ denotes all syntactically valid partial outputs. Formally, if $w_1 \in L_p(G)$ then $\exists w_2 \in \Sigma^*$ such that $w_1.w_2 \in L(G)$*

For a grammar $G$ and a partial output $C_k$ belonging to the set of prefix strings $L_p(G)$, the syntactical decoding problem aims to determine the set $V_k$ of valid tokens from a finite vocabulary $V$ such that appending any token $t \in V_k$ to $C_k$ maintains its syntactic validity according to the grammar $G$.

**Definition 4** (Syntactical Decoding). *For grammar $G$, given partial output $C_k \in L_p(G)$ and finite token vocabulary $V \subset \Sigma^*$, the syntactical decoding problem is to compute the set $V_k \subseteq V$ such that for any $t \in V_k, C_k.t \in L_p(G)$*

We next present SYNCODE's key aspects to solve this problem:

- In the initial step, it parses $C_k$ and computes the unparsed remainder $r \in \Sigma^*$ along with the acceptable terminal sequences $\mathcal{A}$ (Section 4.2).

- In the second step, SYNCODE utilizes $r$, $\mathcal{A}$, and the precomputed mask store. This phase involves traversing the DFA and performing a few lookups within the DFA mask store to obtain the set of syntactically valid tokens $t$ capable of extending $C_k$ (Section 4.3).

- Consequently, SYNCODE efficiently computes the set of syntactically valid tokens. We show the soundness and completeness of our approach in Section 4.4.

- We further discuss the theoretical complexity of SYNCODE in Section 4.6 and the SYNCODE framework in Section 4.7.

## 4.2 Parsing Partial Output

In this section, we describe the remainder $r$ and accept sequences $\mathcal{A}$ returned by the parsing step.

**Remainder.** SYNCODE uses a lexer to convert $C_k$ to sequence of lexical tokens $l_1, l_2 \dots l_f \in \Sigma^*$. Each lexical token $l_i$ is associated with a terminal type $\tau_i$, where $l_i \in L(\rho_{\tau_i})$ ($\rho_{\tau_i}$ is the regular expression for terminal $\tau_i$). We assume our lexer uses a 1-character lookahead without backtracking. This ensures that the lexical types of previous tokens in $C_k$ remain unchanged, except for the final token. The remainder $r$ represents the suffix of $C_k$ that could potentially change its lexical type in future iterations. Thus the remainder $r$ is assigned such that it is either unlexed because it does not match any terminal, or has been lexed but might undergo a different lexing in subsequent iterations when $C_k$ is extended by the LLM by appending tokens. This assumption is crucial for enabling incremental parsing and ensures that the remainder $r$ remains small, which contributes to reducing overall time complexity. SYNCODE assigns the remainder according to the following two cases:

**Case 1:** $C_k = l_1.l_2 \dots l_f$ Assuming a standard lexer with 1-character lookahead and no backtracking, all lexical tokens $l_1, l_2, \dots, l_{f-1}$ remain unchanged upon extending $C_k$. However, the final lexical token $l_f$ may change. For example, in Python partial output in the $k$-th LLM iteration, if the final lexical token is $l_f = \boxed{\texttt{ret}}$ and the language model generates the token $\boxed{\texttt{urn}}$ in the next iteration, the updated code results in the final lexical token becoming $l_f = \boxed{\texttt{return}}$. This transition reflects a transformation from an identifier name to a Python keyword in the subsequent iterations. Thus, $r$ is assigned the value $l_f$, i.e., $r = \boxed{\texttt{ret}}$ for k-th iteration in our example.

**Case 2:** $C_k = l_1.l_2 \dots l_f.u$**:** Here, $u \in \Sigma^*$ is the unlexed remainder of $C_k$. In this case, considering the 1-character lookahead of the lexer, the types of $l_1, l_2, \dots, l_f$ do not change upon extending $C_k$. Consequently, $r$ is assigned value $u$ of the suffix that remains unlexed.

SYNCODE parsing step partitions partial output $C_k$ into lexically fixed part $C_k^{\square}$ and remainder $r$. Given a sequence $\Lambda = \tau_0, \tau_1, \dots, \tau_f$, we simplify notation by using $L(\Lambda) = L(\rho_{\tau_0} \cdot \rho_{\tau_1} \dots \rho_{\tau_f})$ throughout the rest of the paper.

**Definition 5** (Partial Parse). *Given the partial output $C_k \in \Sigma^*$, the partial parse function pparse $: \Sigma^* \to \Gamma^* \times \Sigma^*$ returns a terminal sequence $\Lambda^{\square}$ and remainder $r$ such that $C_k = C_k^{\square}.r$ and $C_k^{\square}$ is parsed as $\Lambda^{\square}$. i.e. $C_k^{\square} \in L(\Lambda^{\square})$.*

**Accept Sequences.** A sentence is a sequence of terminals. A grammar $G$ describes a (possibly infinite) set of sentences, that can be derived by using the production rules of the grammar. We use $L^{\Gamma}(G) \subseteq \Gamma^*$ to denote the valid sequences of terminals that can be derived from the rules of $G$. Further, $L_p^{\Gamma}(G)$ denotes all syntactically valid partial sentences of terminals. Formally,

**Definition 6** (Partial Sentences). *We define a set of all syntactically valid partial sentences $L_p^{\Gamma}(G) \subseteq \Gamma^*$ such that $\Lambda \in L_p^{\Gamma}(G)$ if and only if $\exists \Lambda_1 \in \Gamma^*$ such that $\Lambda.\Lambda_1 \in L^{\Gamma}(G)$.*

Note that $L(G)$ and $L_p(G)$ are defined over alphabet $\Sigma$, whereas $L^{\Gamma}(G)$ and $L_p^{\Gamma}(G)$ over terminals $\Gamma$. Nevertheless, if a program $C$ is parsed to obtain terminal sequence $\Lambda$, then $C \in L(G)$ is equivalent to $\Lambda \in L^{\Gamma}(G)$. The SYNCODE parsing algorithm obtains $\Lambda^{\square} = \tau_1, \tau_2 \dots \tau_f$ by parsing $C_k$ corresponding to the parserd part of partial output $C_k^{\square}$. Given a partial sentence $\Lambda_{\square}$, an accept sequence is a sequence over $\Gamma$ such that when appended to $\Lambda^{\square}$ the result is still a partial sentence.

**Definition 7** (Accept Sequence). *Given partial output $C_k \in L_p(G)$, and $\Lambda^\square, r = pparse(C_k)$, $\Lambda_1 \in \Gamma^*$ is an accept sequence if $\Lambda^\square.\Lambda_1 \in L_p^\Gamma(G)$.*

Consider a Python partial program $C_k = \boxed{\texttt{def is}}$ and let *def*, *name*, *lpar* and *rpar* be the terminals in Python grammar. we get $\{def\}, \boxed{\texttt{is}} = pparse(\boxed{\texttt{def is}})$, where $\Lambda^\square = \{def\}$ and $r = \boxed{\texttt{is}}$. $\Lambda_1 = \{name, lpar, rpar\}$ is an accept sequence in this case as the sequence of terminals $\Lambda^\square.\Lambda_1 = \{def, name, lpar, rpar\}$ is a valid partial sentence. The parser state on parsing the partial output $C_k$ can be utilized to compute a set of accept sequences denoted as $\mathcal{A}$. The soundness and completeness of the SYNCODE algorithm depend on the length of these accept sequences in $\mathcal{A}$. In theory, using longer accept sequences enhances the precision of the SYNCODE algorithm at the cost of increased computational complexity. In Section 4.5, we show our method for obtaining 1 and 2-length accept sequences that are efficient and precise in practice.

### 4.3 Grammar Mask

This section outlines the utilization of the set of acceptable terminal sequences $\mathcal{A}$ and the remainder $r$ in the creation of a boolean mask using the DFA mask store which is subsequently used for constraining the LLM output. The DFA mask store is constructed offline and makes SYNCODE efficient during the LLM generation. Given partial output $C_k$, our objective is to identify tokens $t \in V$ such that appending them to $C_k$ leads to syntactical completion. Given remainder $r$ and set of sequences $\mathcal{A}$, the goal is to determine whether $r.t$ partially matches the regular expression derived from any of the sequences in $\mathcal{A}$. To characterize the notion of strings partially matching a regular expression, we next introduce the function *pmatch*.

**Definition 8** (*pmatch*). *The function pmatch takes a word $w \in \Sigma^*$, a regular expression $\rho$ and returns a boolean. $pmatch(w, \rho) = true$ if either of the following conditions holds:*

    *1. $\exists w_1 \in \Sigma^*, w_2 \in \Sigma^+$ such that $w = w_1.w_2$ and $w_1 \in L(\rho)$ or*

    *2. $\exists w_1 \in \Sigma^*$ such that $w.w_1 \in L(\rho)$*

Thus $pmatch(w, \rho)$ is true when either a prefix of $w$ matches $\rho$ or $w$ can be extended to match $\rho$. The consequence of allowing *pmatch* to be defined such that it is true even when prefix matches, is that SYN-CODE will conservatively accept all tokens for which the prefix matches the accept sequence. Hence, we overapproximate the precise set of syntactically valid tokens. We make this choice to ensure that SYNCODE is sound for any length of accept sequences. Next, we give definitions related to DFAs. These definitions are useful for describing the construction of the DFA mask store and proving properties related to its correctness in the SYNCODE algorithm. In particular, we first define the live states of DFA. We say state $q$ is live if there is a path from $q$ to any final states in $F$. Formally,

**Definition 9** (DFA *live* states). *Given a DFA $D(Q, \Sigma, \delta, q_0, F)$, let $live(Q) \subseteq Q$ denote the set of live states such that*

$$q \in live(Q) \text{ iff } \exists w \in \Sigma^* \text{ s.t. } \delta^*(w, q) \in F$$

We use $D_\tau(Q_\tau, \Sigma_\tau, \delta_\tau, q_0^\tau, F_\tau)$ to denote a DFA corresponding to a terminal $\tau \in \Gamma$. Next, we establish the definition of *dmatch* for DFA, which is an equivalent concept to *pmatch* with regular expressions. *dmatch* is recursively defined such that its computation can be performed by walking over the DFAs of a sequence of terminals.

**Definition 10** (*dmatch*). *Given a DFA $D(Q, \Sigma, \delta, q_0, F)$, a string $w \in \Sigma^*$, a DFA state $q \in Q$ and any sequence of terminals $\Lambda = \{\tau_{f+1}, \tau_{f+2} \ldots \tau_{f+d}\}$, $dmatch(w, q, \Lambda) = true$, if either of the following conditions hold:*

    *1. $\delta^*(w, q) \in live(Q)$ or*

    *2. $\exists w_1 \in \Sigma^*, w_2 \in \Sigma^+$ such that $w_1.w_2 = w$, $\delta^*(w_1, q) \in F$ and $\Lambda = \{\}$ or*

    *3. $\exists w_1 \in \Sigma^*, w_2 \in \Sigma^*$ such that $w_1.w_2 = w$, $\delta^*(w_1, q) \in F$,*
    *and $dmatch(w_2, q_0^{\tau_{f+1}}, \{\tau_{f+2} \ldots \tau_{f+d}\}) = true$ where $q_0^{\tau_{f+1}}$ is the start state corresponding to the DFA for $\tau_{f+1}$*

Given an accept sequence $\Lambda = \{\tau_{f+1}, \tau_{f+2} \ldots \tau_{f+d}\} \in \mathcal{A}$, our objective is to compute the set of tokens $t \in V$ such that $pmatch(r.t, \rho_\Lambda)$ holds, where $\rho_\Lambda = (\rho_{f+1}.\rho_{f+2}.\ldots.\rho_{f+d})$ is the regular expression obtained by concatenating regular expressions for terminals. If $\Lambda^p$ denotes the sequence $\{\tau_{f+2}, \ldots \tau_{f+d}\}$, Lemma 1 simplifies this problem to finding $dmatch(r.t, q_0^{\tau_1}, \Lambda^p)$. Furthermore, utilizing Lemma 2, this can be further reduced to computing $q = \delta_{\tau_1}^*(r, q_0^{\tau_1})$ and $dmatch(t, q, \Lambda^p)$. It's important to note that $dmatch(t, q, \Lambda^p)$ does not depend on $C_k$ and can be computed offline. While the computation of $q$ for $dmatch(t, q, \Lambda^p)$ is relatively inexpensive, evaluating $dmatch(t, q, \Lambda^p)$ can be computationally expensive both offline and online, as it requires considering numerous potential accept sequences offline, and where it needs to iterate over all tokens in $V$ online. We observe that if we consider sequences of smaller lengths, we can efficiently precompute the set of tokens satisfying $dmatch(t, q, \Lambda^p)$ for all $q, t$ and $\Lambda^p$ offline. We later establish the soundness of SYNCODE when using accept sequences of length at least 1 (Theorem 1) and completeness for accept sequences of the length greater than maximum length of tokens in the vocabulary (Theorem 2). Typically, LLM tokens are small in size, allowing us to obtain these guarantees.

**Lemma 1.** *Given* $\Lambda = \{\tau_{f+1}, \tau_{f+2} \ldots \tau_{f+d}\}$, $\Lambda^p = \{\tau_{f+2} \ldots \tau_{f+d}\}$ *and* $\rho_\Lambda = (\rho_{f+1}, \rho_{f+2}, \ldots, \rho_{f+d})$, $dmatch(w, q_0^{\tau_1}, \Lambda^p) \iff pmatch(w, \rho_\Lambda)$.

**Lemma 2.** *If* $q = \delta_\tau^*(r, q_0^\tau)$ *and no prefix of* $r$ *is in* $L(\tau)$ *i.e.* $\nexists w_1 \in \Sigma^*, w_2 \in \Sigma^*$ *such that* $w_1.w_2 = r$ *and* $\delta_\tau^*(w_1, q_0^\tau) \in F_\tau$ *then* $dmatch(t, q, \Lambda) \iff dmatch(r.t, q_0^\tau, \Lambda)$.

The proofs of both the lemmas are in Appendix A.2.

**Illustrative Example:** Consider the scenario with $C_k = \boxed{\texttt{def is}}$, $r = \boxed{\texttt{is}}$, and an accept sequence $\Lambda = \{name, lpar, rpar\}$ in $\mathcal{A}$, where *name*, *lpar*, and *rpar* are terminals in $\Gamma$. Our objective is to determine all $t \in V$ such that $\boxed{\texttt{def is}}.t$ forms a valid partial program. This can be achieved by finding tokens $t$ that satisfy $pmatch(\boxed{\texttt{is}}.t, \rho_\Lambda)$, where $\rho_\Lambda = [a\text{-}z, A\text{-}Z, \_]^*()$. Let's consider a token $t = \boxed{\texttt{\_prime():}}$. We observe that $r.t = \boxed{\texttt{is\_prime():}}$ can be decomposed into $\boxed{\texttt{is\_prime}}$ (*name*), $\boxed{\texttt{(}}$ (*lpar*), $\boxed{\texttt{)}}$ (*rpar*), and $\boxed{\texttt{:}}$. Consequently, it partially matches $\rho_\Lambda$ as defined by *pmatch*. In Figure 9, we present the DFAs for $\Lambda$ used in computing *dmatch*. The reduction $dmatch(r.t, q_0^{name}, lpar, rpar) = dmatch(\boxed{\texttt{is\_prime():}}, q_0^{name}, lpar, rpar)$ simplifies successively to $dmatch(\boxed{\texttt{():}}, q_0^{lpar}, rpar)$, then to $dmatch(\boxed{\texttt{):}}, q_0^{rpar}, )$, and finally to $dmatch(\boxed{\texttt{:}}, q_1^{rpar}, )$. As $q_1^{rpar}$ is a final state, according to condition 2 of Definition 10, $dmatch(\boxed{\texttt{:}}, q_1^{rpar}, )$ holds true. Next, we define a mask over vocabulary

**Definition 11** (Vocabulary mask). *Given vocabulary* $V \subseteq \Sigma^*$, $m \in \{0, 1\}^{|V|}$ *is a mask over the vocabulary. We also use* $set(m) \subseteq V$ *to denote the subset represented by* $m$.

**DFA Mask Store** For an integer $\alpha$, we define a DFA table $\mathcal{M}_\alpha$ as the mask store over the DFA states with $\alpha$ lookahead. Given the set of all DFA states $Q_\Omega = \bigcup_{\tau \in \Gamma} Q_\tau$, the table stores binary masks of size $|V|$, indicating for token string $t$, for any DFA state $q \in Q_\Omega$ and a sequence of $\alpha$ terminals $\Lambda_\alpha$ if $dmatch(t, q, \Lambda_\alpha) = true$. The lookahead parameter $\alpha$ signifies the number of subsequent terminals considered when generating the mask stored in the table. Choosing a larger value for $\alpha$ enhances the precision of SYNCODE algorithm, but it comes at the cost of computing and storing a larger table. We next formally define the DFA mask store,

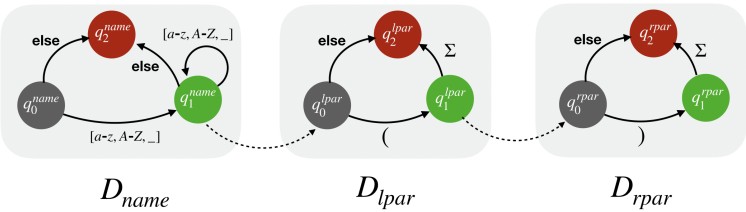

Figure 8: DFAs in accept sequence $\Lambda = \{name, lpar, rpar\}$ for example. The start state, final states, and dead states are in gray, green, and red respectively. The dashed arrows link the final states of one DFA to the starting state of the next DFA, adhering to condition 3 in Definition 10. This illustrates the sequential traversal across DFAs during the computation of *dmatch*.

**Definition 12** (DFA mask store). *For an integer $\alpha$, the DFA mask store $\mathcal{M}_\alpha$ is a function defined as $\mathcal{M}_\alpha : Q_\Omega \times \Gamma^\alpha \to \{0,1\}^{|V|}$, where $Q_\Omega = \bigcup_{\tau \in \Gamma} Q_\tau$ represents the set of all DFA states and $\Gamma^\alpha$ is a set of $\alpha$-length terminal sequences. Then $\mathcal{M}_\alpha(q, \Lambda) = m$ is a binary mask such that $t \in set(m)$ if $dmatch(t, q, \Lambda)$*

For our illustrative example if $m = \mathcal{M}_2(q_1^{name}, \{lpar, rpar\})$ then $t = \boxed{\texttt{\_prime():}}$ should be contained in $set(m)$. The grammar mask for a set of accept sequences $\mathcal{A}$ can be computed by combining masks for each $\Lambda \in \mathcal{A}$. The DFA mask store $\mathcal{M}_0$ maps each DFA state to all tokens such that they *pmatch* without considering any following accept sequence (0-length sequence). In this case, the table maps each state with a single mask denoting the tokens that match the regular expression of the corresponding DFA.

**Computing Grammar Mask** The mask store is constructed offline by enumerating all DFA states $Q_\Omega$, considering all possible terminals in $\Gamma$, and all tokens in $V$. The DFA mask store depends on the set of terminals $\Gamma$ and the model's vocabulary $V$. As a result, a unique mask store is created for each grammar and tokenizer combination, and to enhance efficiency, we cache and reuse this table for future inferences.

Algorithm 2 presents our approach for computing the grammar mask during LLM generation. It computes a grammar mask based on the sets of current accept sequences $\mathcal{A}$, and the remainder string ($r$). It iterates over $\mathcal{A}$, considering each sequence $\Lambda$. The algorithm initializes an empty mask $m$. It iterates over each acceptable sequence, considering the first terminal $\tau_1$ in each. It computes the resulting state $q_r$ by processing $\tau_1$ from an initial state $q_0^{\tau_1}$ and the remainder string $r$. If $q_r$ is in a live state, the algorithm updates the grammar mask by unifying the mask cached in $\mathcal{M}_\alpha$.

---

**Algorithm 2** Computing Grammar Mask

**Inputs:** $\mathcal{A}$: set of accept sequences, $r$: remainder

1: **function** GRAMMARMASK($\mathcal{A}, r$)
2: $\quad m \leftarrow \{\}$
3: $\quad$ **for** $\Lambda \in \mathcal{A}$ **do**
4: $\quad\quad \tau_1 \leftarrow \Lambda[0]$
5: $\quad\quad q_r \leftarrow \delta^*(q_0^{\tau_1}, r)$
6: $\quad\quad$ **if** $q_r \in live(Q_{\tau_1})$ **then**
7: $\quad\quad\quad \Pi \leftarrow len(\Lambda) - 1$
8: $\quad\quad\quad m \leftarrow m \cup \big(\mathcal{M}_\Pi(q_r, \Lambda[1:])\big)$
9: $\quad$ **return** $m$

---

### 4.4 Soundness and Completeness

This section establishes the soundness and completeness of the SYNCODE algorithm. Algorithm 3 presents the LLM generation algorithm with SYNCODE. It takes as inputs an LLM represented by $M$, a tokenizer denoted by $\mathcal{T}$, an input prompt string $C_0$, the maximum number of generated tokens $n_{max}$, and a base decoding strategy $D$. The algorithm begins by tokenizing the input prompt using the tokenizer. It then iteratively generates tokens using the LLM, decodes the current token sequence, and performs parsing to obtain acceptable terminal sequences $\mathcal{A}$, and a remainder $r$ (line 6). A grammar mask is applied to the logit scores based on these values (line 7). The algorithm subsequently selects the next token using the decoding strategy, and if the end-of-sequence token (EOS) is encountered, the process terminates. The final decoded output is obtained, incorporating the generated tokens, and is returned as the result of the MaskedGenerate algorithm.

Given partial output $C_k \in L_p(G)$, SYNCODE generates a corresponding mask $m$. If, for a token $t \in V$, the concatenation $C_k.t$ results in a syntactically valid partial output, i.e. $C_k.t \in L_p(G)$, our soundness theorem ensures that $t$ is indeed a member of the set defined by the generated mask $m$. The subsequent theorem formally states this soundness property.

**Theorem 1.** *Let $C_k \in L_p(G)$ be the partial output and any integer $d \geq 1$, let $\mathcal{A}_d \subseteq \Gamma^d$ contain all possible accept terminal sequences of length $d$ and $r \in \Sigma^*$ denote the remainder. If $m = GrammarMask(\mathcal{A}, r)$ then for any $t \in V$, if $C_k.t \in L_p(G)$ then $t \in set(m)$*

The proof of the theorem is in Appendix A.2.

Next, we give a definition that establishes a partial order on sets of terminal sequences, where given two sets $\mathcal{A}_1$ and $\mathcal{A}_2$, we say sets $\mathcal{A}_1 \preccurlyeq \mathcal{A}_2$ if every sequence in $\mathcal{A}_2$ has a prefix in $\mathcal{A}_1$.

**Definition 13** ($\preccurlyeq$). *We define a partial order $\preccurlyeq$ on set of terminal sequences $\mathcal{P}(\Gamma^*)$ such that $\mathcal{A}_1 \preccurlyeq \mathcal{A}_2$ when $\forall \Lambda_2 \in \mathcal{A}_2 \exists \Lambda_1 \in \mathcal{A}_1 \exists \Lambda_3 \in \Gamma^* \ s.t. \ \Lambda_2 = \Lambda_1.\Lambda_3$*

We further state the lemma that shows the relation in the grammar masks generated by two accept sequences satisfying relation $\preccurlyeq$.

**Lemma 3.** *Given $\mathcal{A}_1$ and $\mathcal{A}_2$ are set of accept sequences such that $\mathcal{A}_1 \preccurlyeq \mathcal{A}_2$ and $m_1 = GrammarMask(\mathcal{A}_1, r)$ and $m_2 = GrammarMask(\mathcal{A}_2, r)$ then $set(m_2) \subseteq set(m_1)$*

The proof of the lemma is in Appendix A.2.

Theorem 1 proves soundness for accept sequences $\mathcal{A}_d$ of length $d$, while Lemma 3 extends this proof to any set of accept sequences $\mathcal{A}$ where $\mathcal{A} \preccurlyeq \mathcal{A}_d$. Our implementation, employing sequences of varying lengths, can be proven sound based on this extension.

The completeness theorem ensures that, under specified conditions, each token $t \in set(m)$ guarantees $C_k.t$ as a syntactically valid partial output. An implementation of SYNCODE with a short length of

---

**Algorithm 3** SYNCODE Generation

**Inputs:** $M$: LLM, $\mathcal{T}$: tokenizer, $C_0$: input prompt, $n_{max}$: maximum generated tokens, $D$: decoding strategy

1: **function** MASKEDGENERATE($M$, $\mathcal{T}$, $C_0$, $n_{max}$, $D$)
2:     $T_{cur} \leftarrow$ Tokenize($\mathcal{T}, C_0$)
3:     **for** $i \in \{1, \ldots n_{max}\}$ **do**
4:         $scores \leftarrow M(T_{cur})$
5:         $C_k \leftarrow$ decode($\mathcal{T}, T_{cur}$)
6:         $\mathcal{A}, r \leftarrow$ Parse($C_k$)
7:         $m \leftarrow$ GrammarMask($\mathcal{A}, r$)
8:         $scores \leftarrow m \odot scores$
9:         $t_i \leftarrow D(scores)$
10:       **if** $t_i = EOS$ **then**
11:          break
12:       $T_{cur} \leftarrow$ append($T_{cur}, t_i$)
13:     output $\leftarrow$ decode($\mathcal{T}, T_{cur}$)
14:     **return** output

---

accept sequences although sound, may not guarantee completeness. To illustrate, let's take the example where $\Lambda = \tau_{f+1}, \tau_{f+2} \in \mathcal{A}$ with simple singleton regular expressions $\rho_{\tau_{f+1}} = \boxed{\mathtt{(}}$ and $\rho_{\tau_{f+2}} = \boxed{\mathtt{(}}$. In this case, our algorithm conservatively treats all tokens $t \in V$ as syntactically valid, whenever $\boxed{\mathtt{((}}$ is a prefix of those tokens (e.g., $\boxed{\mathtt{(((}}$, $\boxed{\mathtt{(()}}$)) even though some tokens may not meet syntactic validity. However, by assuming that the accept sequences are long enough, we can establish the completeness of the approach.

**Theorem 2.** *Let $C_k \in L_p(G)$ be the partial output, let $\mathcal{A}_d \subseteq \Gamma^d$ contain all possible accept terminal sequences of length $d$ and $r \in \Sigma^*$ denote the remainder. Suppose for any $t \in V, d > len(t)$ and $m = GrammarMask(\mathcal{A}_d, r)$ such that $t \in set(m)$ then $C_k.t \in L_p(G)$*

The proof of the theorem is in Appendix A.2. While our completeness theorem ensures the SYNCODE consistently extends syntactically correct partial outputs, it does not guarantee termination with a correct and complete output. The focus of the theorem is on generating syntactically valid partial outputs, and the theorem does not address whether the process converges to a syntactically correct whole output. Termination considerations go beyond the completeness theorem's scope.

## 4.5 SynCode Implementation

**Base LR parser:** Bottom-up LR parsers, including LR(1) and LALR(1) parsers, process terminals generated from the lexical analysis of the code sequentially and perform shift or reduce operations (Aho et al., 1986). LR($\kappa$) parsers have the immediate error detection property, ensuring they do not perform shift or reduce operations if the next input $\kappa$ terminals on the input tape is erroneous (Aho and Johnson, 1974). Consequently, every entry in the parsing table corresponding to $\kappa$ terminals that maps to a shift or reduce operation indicates that the terminal is acceptable. This property allows us to use LR(1) parsing tables to efficiently compute accept sequences at any intermediate point, making them preferable for SYNCODE applications. Thus, computing acceptable terminals with LR(1) parsers has a complexity of $O(|\Gamma|)$. Although LALR(1) parsers are more commonly used due to their smaller memory requirements and faster construction, computing acceptable terminals with them requires iterating over all terminals leading to a complexity of $O(T_P \cdot |\Gamma|)$ due to the need for multiple reduce operations before confirming the validity of each terminal. Furthermore, while for $\kappa > 1$, LR($\kappa$) parsers can compute accept sequences of length $\kappa$ immediately, they incur extremely high memory requirements. Additionally, while we can use LL($\kappa$) parsing tables to compute the next $\kappa$ accept terminals, LR($\kappa$) parsers offer a higher degree of parsing power. Therefore, we employ LR parsers in SYNCODE. Our evaluation indicates that LR(1) parsers suffice for eliminating most syntax errors, making them a practical choice for SYNCODE. We discuss how the implementation of how parsing is performed *incrementally* to obtain the accept sequences and remainder in the Appendix A.3.

**Accept Sequences:** In our implementation, we focus on generating accept sequences of length 1 or 2, as they can be efficiently obtained from LR(1) parser. While this approach incurs some loss of precision, it leads to sound but incomplete syntactical decoding. Further, our evaluation demonstrates that this strategy is efficient and precise in practical scenarios. We note that *pmatch r.t* with a 2-length sequence is equivalent to *dmatch* with a 1-length sequence, as stated in Lemma 1. Consequently, in our work, we precompute mask stores $\mathcal{M}_0$ and $\mathcal{M}_1$. On parsing the partial output $C_k$, the parser state of LR(1) parsers can be used to directly obtain syntactically acceptable terminals for the current completion ($A_0$) and the next completion ($A_1$). We utilize $A_0$ and $A_1$ to construct the accept sequences $\mathcal{A}$, considering two cases:

**Case 1:** $C_k = l_1.l_2 \ldots l_f$**:** Let $\tau_f$ represent the type of the final lexical token. In many instances, a token may be extended in the subsequent generation step, such as when an identifier name grows longer or additional words are appended to a comment. In those cases if $A_1 = \tau_1^1, \tau_2^1, \ldots, \tau_n^1$, we include all 2-length sequences $\{\tau_f, \tau_i^1\}$ for each $i$. As previously discussed, the type of the final lexical token may change from $\tau_f$. Consequently, when $A_0 = \{\tau_1^0, \tau_2^0, \ldots, \tau_n^0\}$, we add 1-length sequences $\Lambda_i$ for each terminal sequence $\{\tau_i\}$ from $A_0$, excluding $\tau_f$. This method ensures the generation of sequences accounting for potential extensions of the same token and changes in the type of the final lexical token.

**Case 2** $C_k = l_1.l_2 \ldots l_f.u$**:** In this scenario, the current terminal is incomplete, leading to a lack of information about subsequent terminals. Consequently, when $A_1 = \{\tau_1, \tau_2, \ldots, \tau_n\}$, we define $\mathcal{A}$ as a set of sequences: $\{\Lambda_1, \Lambda_2, \ldots, \Lambda_n\}$, where each $\Lambda_i$ corresponds to a single terminal sequence $\{\tau_i\}$ from $A_1$. Specifically, $\Lambda_1 = \{\tau_1\}$, $\Lambda_2 = \{\tau_2\}$, and so forth.

### 4.6 Time and Space Complexity

In this section, we analyze the time complexity of the SYNCODE algorithm. We focus on the cost of creating the mask at each iteration of the LLM generation loop. The key computations involved in this process are the parsing carried out by the incremental parser to compute $\mathcal{A}$ and the lookup/unification operations performed through the DFA mask store.

The incremental parser parses $O(1)$ new tokens at each iteration and computes $\mathcal{A}$. Let $T_A$ represent the time taken by the parser to compute the accepted terminals and $T_P$ denote the time the parser takes to parse a new token and update the parser state. Hence, in each iteration, the parser consumes $O(T_A + T_P)$ time to generate $\mathcal{A}$. The DFA mask store lookup involves traversing $|\mathcal{A}|$ DFA sequences, with the number of steps in this walk bounded by the length of the remainder $r$. As $\mathcal{A}$ can have a maximum of $|\Gamma|$ sequences, the DFA walk consumes $O(|\Gamma| \cdot len(r))$ time. We employ a hashmap to facilitate efficient lookups at each DFA node, ensuring that all lookups take constant time. Consequently, this step takes $O(|\Gamma|)$ time. Let $T_\cup$ denote the time taken for computing the union of binary masks. With potentially $|\Gamma|$ union operations to be performed, the mask computation takes $O(T_\cup \cdot |\Gamma|)$ time. Therefore, the overall time complexity at each step during generation is given by $O(T_A + T_P + |\Gamma| \cdot len(r) + T_\cup \cdot |\Gamma|)$.

For an incremental LR(1) parser, the complexity of our algorithm at each step of LLM token generation is $O(|\Gamma| \cdot len(r) + T_\cup \cdot |\Gamma|)$. With our lexer assumption, we ensure that the remainder $r$ is small, allowing us to further simplify our complexity analysis to $O(T_\cup \cdot |\Gamma|)$ by treating $len(r)$ as constant.

**Offline cost:** The cost of computing the mask store $\mathcal{M}_\alpha$ offline involves considering all DFA states $q \in Q_\Omega$, all possible terminal sequences of length $\alpha$, and all tokens $t \in V$. Given that we need to traverse the DFA for $len(t)$ steps for each entry in the store, the time complexity for computing the mask store is $O(max_{t \in V}(len(t)).|Q_\Omega|.|V|.|\Gamma|^\alpha)$. Typically, $len(t)$ is small, allowing us to simplify this to $O(|Q_\Omega|.|V|.|\Gamma|^\alpha)$. In our implementation, the use of $\mathcal{M}_0$ and $\mathcal{M}_1$ results in a cost of $O(|Q_\Omega|.|V|.|\Gamma|)$. The size of $|Q_\Omega|$ depends on the complexity of regular expressions for the terminals, which may vary for each grammar. However, as demonstrated in our evaluation section, these mask stores can be computed within 10 minutes for each combination of grammar and LLM. This computation is a one-time cost that can be amortized over all generations performed for the given LLM and grammar.

**Space complexity:** The mask store $\mathcal{M}_\alpha$ consists of masks of size $|V|$ for each DFA state $q \in Q_\Omega$ and for all possible terminal sequences of length $\alpha$. Consequently, the size of the mask store is $O(|Q_\Omega| \cdot |V| \cdot |\Gamma|^\alpha)$.

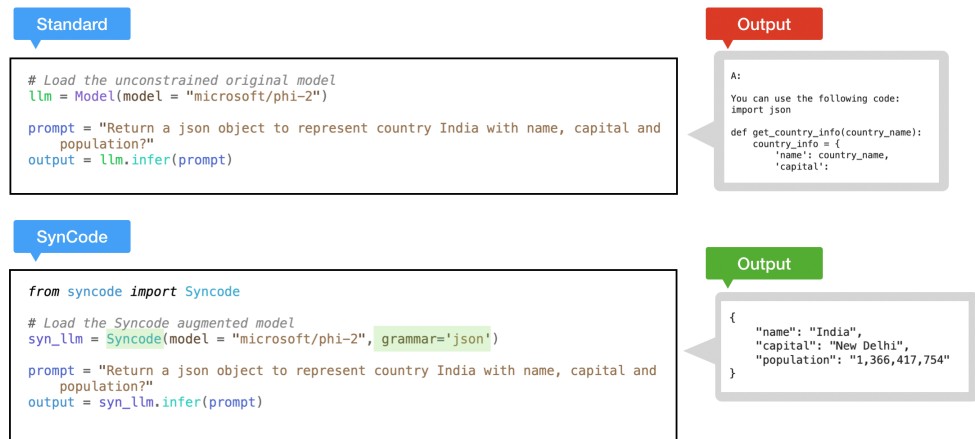

Figure 9: The upper section displays erroneous output from a standard LLM generation, failing to produce the intended JSON format. The lower segment showcases the fix achieved through the use of the SynCode framework.

In our implementation, the use of $\mathcal{M}_0$ and $\mathcal{M}_1$ reduces the size to $O(|Q_\Omega| \cdot |V| \cdot |\Gamma|)$. In our evaluation, the size of the mask store is approximately 1–2 GB, as shown in Table 6.

### 4.7 SynCode Framework

Figure 9 shows how SynCode framework can be used in practice by selecting a grammar. We next discuss other important features of the framework.

**Adding a New Grammar.** Our Python-based SynCode framework is shipped with several built-in grammars such as JSON, Python, Go, etc. A user can apply SynCode for arbitrary grammar by providing the grammar rules in EBNF syntax with little effort. The grammar needs to be unambiguous LALR(1) or LR(1) grammar for using the respective base parsers.

**Ignore Terminals.** Our EBNF syntax adopted from Lark allows one to provide *ignore terminals* as part of the grammar. Lark ignores those terminals while parsing. In the case of Python, this includes *comments* and *whitespaces*. SynCode handles these ignore terminals by adding a trivial 1-length accept sequence for each of these ignore terminals.

**Parsers.** SynCode supports both LALR(1) and LR(1) as base parsers. We adapt Lark's (Lark, ) LALR(1) parser generator for SynCode. Since Lark does not implement the LR(1) parser generator, we implemented the LR(1) parser generator on top of the Lark. The generation of LR(1) parser which is performed offline may take longer time compared to the LALR(1) parser (e.g., up to 2 mins for our Python grammar), however, it is more efficient at inference time in computing the accept sequences. Further, since the Lark-generated parser is non-incremental, we build the incremental parser on top of it by caching the parser state as described in Appendix A.3.

**Non-CFG Fragments of PLs.** SynCode can handle non-context-free fragments of PLs, such as *indentation* in Python and end-of-scope markers in Go. To support languages with indentation, such as Python and YAML, SynCode has a mechanism that tracks acceptable indentation for the next token, effectively masking tokens that violate indentation constraints at a given point. This indentation constraint feature can be enabled with any new grammar. Similarly, for handling other custom parsing rules beyond CFGs, users can add additional constraints to the generation by overriding specific SynCode functions. For instance, in Go, semicolons are optional and may be automatically inserted at the end of non-blank lines. Implementing such constraints in SynCode programmatically requires minimal effort. However, SynCode currently does not support enforcing semantic constraints. (e.g, if a variable in a program is defined before it is used.)

## 5   Experimental Methodology

**Models.** In our evaluation, we select a diverse set of state-of-the-art open-weight LLMs of varying sizes. Since closed-source LLMs, such as GPT-4 or Gemini, do not expose generation logits through their APIs, applying a constrained generation approach in SYNCODE is not feasible. Therefore, we focus on enhancing smaller, open-source models in our evaluation. We select the state-of-the-art models Llama-2-7B-chat (Touvron et al., 2023b) and Gemma2-2B-it  (Team et al., 2024) for our JSON evaluation. For text-2-SQL generation experiments, we use Llama-2-7B-chat, Llama-3.2-1B, Llama-3.2-3B, and Gemma-2-2B-it.  Furthermore, we chose models such as LLaMA-7B (Touvron et al., 2023a), WizardCoder-1B (Luo et al., 2023), and CodeGen-350M (Nijkamp et al., 2023) for code completion.

**Datasets.** We focus our evaluation on generating JSON, SQL, Python, and Go outputs. We choose JSON as it is supported by the baselines (Gerganov and et. al., 2024; Willard and Louf, 2023), which allows us to compare against them. We selected Python since it is extensively present in the training data employed for LLM training and fine-tuning. Conversely, we opted for Go due to its lower standard LLM accuracy and a relatively smaller presence in the training data. We consider JSON-Mode-Eval (NousResearch, 2024) dataset for text to JSON generation and HumanEval and MBXP (Athiwaratkun et al., 2023) dataset for evaluating Python and Go code generation. We display examples of prompts from these datasets in Appendix A.7.

- **JSON-Mode-Eval (NousResearch, 2024).** It consists of 100 zero-shot problems. Each problem prompt follows the chat format with a system prompt specifying a JSON schema and a user prompt requesting the LLM to generate a JSON object that contains specified contents.

- **Spider text-2-SQL.** Spider (Yu et al., 2018) text-to-SQL dataset consists of 1,034 problems of varying difficulty levels: *easy* (250), *medium* (440), *hard* (174), and *extra hard* (170).

- **Multilingual HumanEval (Athiwaratkun et al., 2023).** It is an extension of the original HumanEval collection (Chen et al., 2021), which comprises 164 Python programming problems, to include other languages like Go. Each problem in the dataset consists of a function definition, and text descriptions of the function as a part of the function docstring.

- **MBXP (Athiwaratkun et al., 2023).** It is extended from the MBPP (Austin et al., 2021) dataset for Python to support other languages such as Go. The dataset consists of 974 problems with the same format as HumanEval.

**Grammars.** For Python, we used the readily available grammar from the Lark repository. For Go, we converted an existing LL(*) grammar from (ANTLR,  ) implementation to LR(1) grammar for our use. We write the CFG for these languages using the Extended Backus-Naur Form (EBNF) syntax. We use a substantial subset of grammar for Python and Go syntactic generation with SYNCODE. The grammar has commonly used features of the language such as control flow, and loops, and excludes some features such as Python's support for lambda functions. Adding support for more features would require more engineering effort but it will not change the overall technique. The grammars we used are available in Appendix A.8. The JSON grammar consists of 19 rules and 12 terminals. The Python grammar we used contains 520 production rules and 94 terminals, whereas the Go grammar comprises 349 rules and 87 terminals.

**Evaluating Syntax Errors.** For evaluating the errors in the generated output in each of the languages, we use their respective standard compilers.

**Experimental Setup.** We run experiments on a 48-core Intel Xeon Silver 4214R CPU with 2 NVidia RTX A5000 GPUs. SYNCODE is implemented using PyTorch (Paszke et al., 2019a), HuggingFace transformers library (Wolf et al., 2020) and Lark library (Lark,  ).

**Baselines.** We evaluate three state-of-the-art baselines OUTLINES (Willard and Louf, 2023) v0.1.1, GUIDANCE (Lundberg et al., 2023) v0.1.16 and GCD (Geng et al., 2023) in our study. The algorithmic differences in the baselines and SYNCODE are discussed in Section 7. We perform a warmup run for each experiment where we measure inference time to ensure that one-time precomputation time is not included in the inference runtime. For a fair comparison with baselines, SYNCODE uses opportunistic masking (Beurer-Kellner

et al., 2024), an optimization used in LLAMA.CPP and GUIDANCE. Instead of computing the full logit vector mask upfront, the model generates a token and only computes the mask if the proposed token is incorrect.

## 6 Experimental Results

In this section, we evaluate SYNCODE on generating various formal languages. We compare SYNCODE with state-of-the-art baselines and perform various ablation studies.

SYNCODE allows the model to generate a special EOS token (indicating the end of generation) only when the output belongs to $L(G)$. In practice, however, LLM generation typically stopped after a fixed maximum number of tokens, $n_{max}$. Therefore, terminating with the EOS token within this limit is not always guaranteed potentially resulting in syntax errors.

### 6.1 Effectiveness of SynCode for JSON Generation

Table 1: Effectiveness of SYNCODE in generating JSON with original and explicit prompts. Column generation time (with standard deviation of the mean) in seconds.

| Model | Tool | Syntax Errors | | Validation Accuracy (%) | | Generation Time (s) | |
|---|---|---|---|---|---|---|---|
| | | Original | Explicit | Original | Explicit | Original | Explicit |
| Llama-2-7B-chat | **SynCode** | **0** | **0** | **66%** | **84%** | **3.08** $_{\pm 0.1}$ | **3.04** $_{\pm 0.06}$ |
| | Standard | 98 | 41 | 2% | 58% | 3.61 $_{\pm 0.09}$ | 3.15 $_{\pm 0.09}$ |
| | GUIDANCE | 13 | 11 | 57% | 65% | 5.14 $_{\pm 0.16}$ | 4.22 $_{\pm 0.08}$ |
| | OUTLINES[†] | 16 | 14 | 62% | 56% | 38.07 $_{\pm 0.27}$ | 41.79 $_{\pm 0.2}$ |
| | GCD | 2 | 0 | 62% | 64% | 6.08 $_{\pm 0.2}$ | 4.01 $_{\pm 0.19}$ |
| Gemma2-2B-it | **SynCode** | **0** | **0** | **99%** | **100%** | 4.84 $_{\pm 0.09}$ | 4.7 $_{\pm 0.05}$ |
| | Standard | 59 | 59 | 41% | 41% | 4.32 $_{\pm 0.16}$ | 5.82 $_{\pm 0.06}$ |
| | GUIDANCE | 1 | 1 | 96% | 96% | 6.09 $_{\pm 0.29}$ | 5.56 $_{\pm 0.24}$ |
| | OUTLINES | 2 | 0 | 67% | 90% | **1.99** $_{\pm 0.13}$ | **2.75** $_{\pm 0.32}$ |
| | GCD | 1 | 0 | 96% | 95% | 19.12 $_{\pm 0.08}$ | 8.82 $_{\pm 0.13}$ |

† We observed issues when using Llama-2-7B-chat with Outlines v0.1.1 and therefore, we use older version v0.0.46.

We evaluate the effectiveness of SYNCODE in guiding LLMs with the JSON grammar to generate syntactically correct JSON. We run the inference with Llama-2-7B-chat and Gemma2-2B-it with SYNCODE, OUTLINES, GUIDANCE, GCD, and standard generation on the 100 problems from the JSON-Mode-Eval dataset. We select these models for the JSON experiment as they are supported by all considered baselines.

We set max new tokens $n_{max} = 400$. We also report an evaluation of augmenting the prompts with an explicit request to output only JSON. We present an example of these explicit prompts in Appendix A.7. We evaluate the correctness of JSON generated by an LLM by first evaluating whether the JSON string can be parsed and converted to a valid JSON object. We further evaluate whether the generated JSON is valid against the schema specified in the prompt. Although the SYNCODE does not enforce the specific schema to the JSON output for each task, we believe it is an important research question to check whether the reduced syntax errors due to SYNCODE can also lead to improved schema validity.

Table 1 presents our evaluation results. We report results for both the prompts taken directly from the dataset (denoted as "Original") and after augmenting these prompts with an explicit request to output JSON (denoted as "Explicit"). In the "Validation Accuracy" column, we compute the percentage of valid completions against their respective schemas. In the "Generation Time (s)" column, we report the average time taken to generate a completion to a prompt from the dataset. Guiding Llama-2-7B-chat and Gemma2-2B-it with the JSON grammar via SYNCODE eliminates syntax errors in generated JSON. On the other hand, standard generation results in syntactically incorrect JSON for 98% and 59% of completions to the original prompts for the Llama-2-7B-chat and Gemma2-2B-it models respectively. A majority of these errors are due to the generation of natural language before and after the JSON. Explicit prompts somewhat mitigate this issue, but still results in syntactically invalid outputs to 41% and 59% of these prompts for standard Llama-2-7B-chat and Gemma2-2B-it generation respectively, primarily due to errors such as unmatched braces and unterminated string literals. OUTLINES, GUIDANCE, and GCD face similar problems with closing braces and terminating strings.

Notably, SYNCODE significantly improves the JSON schema validation accuracy of Gemma2-2B-it completions over standard generation, from 41% to 99% and 41% to 100% for original and explicit prompts respectively. Furthermore, SYNCODE outperforms OUTLINES,GUIDANCE, and GCD in validation accuracy of Llama-2-7B-chat completions by 4%, 9%, and 4% respectively for original prompts and 28%, 19%, and 20% for explicit prompts. The remaining schema validation errors with SYNCODE are semantic errors, including data type mismatch between the generation JSON and schema, missing fields required by the schema, and adding extra fields not allowed by the schema. SYNCODE is faster than all baseline grammar-guided generation methods for Llama-2-7B-chat and all but OUTLINES for Gemma2-2B-it. The low generation time with OUTLINES for Llama-2-7B-chat can largely be attributed to the fact that many of its completions to prompts are empty JSON (35% of original and 7% of explicit) which takes few tokens to generate but often does not conform to the schema.

Interestingly, we observe that for Llama-2-7B-chat, SYNCODE also reduces the average generation time over standard generation. We attribute this finding to the fact that without grammar-guided generation, the model generates syntactically invalid output, such as natural language, in addition to JSON and thus generates more tokens in response to the same prompt than with SYNCODE. Thus, augmenting LLMs with SYNCODE can significantly improve syntactical correctness and runtime efficiency.

## 6.2 Effectiveness of SynCode for SQL Generation

This study demonstrates that SYNCODE improves text-to-SQL generation by enforcing grammar constraints, ensuring that generated SQL queries are syntactically accurate. We measure the execution accuracy of generated SQL queries using tests provided in Spider (Yu et al., 2018) benchmark. We evaluate the following models for SQL generation: Llama-3.2-1B, Llama-3.2-3B (base models) and Llama-2-7B-chat, Gemma-2-2B-it (instruct-tuned models). We observe that despite explicitly prompting to only generate the SQL query, the instruct-tuned Gemma-2-2B-it model often enclosed generated SQL queries within markers, such as ` ``` ` or ` ```sql `. Thus, we consider another baseline for Gemma-2-2B where we extract the SQL query substring within these markers, handling cases where the output format is either ` ```{SQL query} ``` ` or ` ```sql {SQL query} ``` `.

For evaluation, we use the Spider (Yu et al., 2018) text-to-SQL dataset, which consists of 1,034 problems of varying difficulty levels: *easy* (250), *medium* (440), *hard* (174), and *extra hard* (170). We prompt models with schema information and text queries, instructing them to generate SQL queries only. Using greedy decoding and `\n\n` is used as an additional stopping condition for all experiments.

Table 2: Comparison of SYNCODE and unconstrained generation on SQL generation.

| Model | Method | Accuracy (%) | | | | | Execute (%) | Tokens | Time (s) |
|---|---|---|---|---|---|---|---|---|---|
| | | Easy | Medium | Hard | Extra | Overall | | | |
| Gemma2-2B-it | Standard | 0.0 | 0.0 | 0.0 | 0.0 | 0.0 | 0.0 | 221.43 | 9.883 |
| | Standard+ | 44.8 | 18.9 | 21.9 | 17.1 | 25.4 | 77.6 | 221.43 | 9.893 |
| | SYNCODE | **45.8** | 18.7 | **23.3** | **20.6** | **26.3** | **78.2** | **135.17** | **5.876** |
| Llama-2-7b-chat | Standard | 34.4 | 22.0 | 12.1 | 4.1 | 20.4 | 32.6 | 44.74 | 1.148 |
| | SYNCODE | **40.0** | **27.3** | **13.8** | **5.9** | **24.6** | **41.6** | 50.33 | 1.483 |
| Llama-3.2-1B | Standard | 40.8 | 24.8 | 20.7 | 10.6 | 25.6 | 51.1 | 48.00 | 0.509 |
| | SYNCODE | **46.8** | **28.2** | **23.0** | 10.6 | **28.8** | **59.0** | 56.36 | 0.916 |
| Llama-3.2-3B | Standard | 38.0 | 29.5 | 28.2 | 12.9 | 28.6 | 67.4 | 47.78 | 0.846 |
| | SYNCODE | **47.2** | **34.8** | **32.8** | **19.4** | **34.9** | **81.4** | 47.63 | 1.164 |

Table 2 presents a comparison of SYNCODE and unconstrained generation across key metrics. The Accuracy (%) column shows the percentage of correctly generated SQL queries across different difficulty levels. Execute (%) reflects the percentage of queries successfully executed without runtime errors in SQLite. The Tokens column reports the average number of tokens generated, and Time(s) shows the average generation time. Standard+ row for Gemma2 denotes the result for the additional baseline where we extract the SQL query from the full generation using regex matching.

Table 3: Comparison of SynCode with Outlines generation on SQL generation.

| Model | Outlines | | SynCode | |
|---|---|---|---|---|
| | **Accuracy** | **Avg Time (s)** | **Accuracy** | **Avg Time (s)** |
| Llama-3.2-1B | 0.04 | 7.88 | 0.25 | 0.70 |
| Llama-3.2-3B | 0.17 | 11.16 | 0.31 | 1.03 |
| Gemma-2-2b-it | 0.21 | 45.82 | 0.25 | 5.81 |

We observe that SynCode achieves better performance over the baselines in terms of both execution percentage and execution accuracy. For example, with the Llama-3.2-3B model, SynCode achieves an execution success rate of 81.4%, compared to 67.4% for unconstrained generation. Further, the execution accuracy improves from 28.6% to 34.9%. In the case of the Gemma2-2B-it model, we observe that SynCode shows a moderate improvement over the Standard+ accuracy. However, it shows a significant gain in the speed (1.7x) of generation and a reduction in the number of tokens generated. Although the Gemma2-2B-it model has a good execution percentage without any runtime errors. The instruct-tuned models tends to use large number of tokens that are not part of the query. In applications where the goal is to use LLMs to generate SQL queries without additional explanations, the result with Gemma2-2B-it shows that SynCode is useful in improving the efficiency of LLM generation along with the improvements in accuracy.

**Comparison to Outlines:** We perform additional experiment on the first 100 exampels in the Spider dataset on the task of SQL generation. Table 3 presents the accuracy and Avg. time taken for various models. Our results show that SynCode is significantly faster than Outlines in SQL generation.

## 6.3 Effectiveness of SynCode for GPL

Table 4: Number of programs with syntax errors for standard and SynCode generation ($\downarrow$ shows how much SynCode reduces the occurrence of the syntax errors compared to Standard generation.

| Dataset | Model | Python | | | Go | | |
|---|---|---|---|---|---|---|---|
| | | Standard | SynCode | $\downarrow$ | Standard | SynCode | $\downarrow$ |
| HumanEval | CodeGen-350M | 271 | **15** | 95% | 573 | **49** | 91% |
| | WizardCoder-1B | 36 | **3** | 92% | 1031 | **50** | 95% |
| | LLaMA-7B | 291 | **2** | 99% | 725 | **10** | 99% |
| MBXP | CodeGen-350M | 78 | **4** | 95% | 212 | **2** | 99% |
| | WizardCoder-1B | 28 | **2** | 93% | 243 | **14** | 94% |
| | LLaMA-7B | 148 | **5** | 97% | 414 | **1** | 99% |

We run inference with CodeGen-350M, WizardCoder-1B, and LLaMA-7B with SynCode and with the standard no-masking approch. We do not compare SynCode with the other baselines as none of these works support general-purpose programming language grammars. We experiment with both Python and Go programming languages, evaluating performance on zero-shot problems from the HumanEval and MBXP datasets. For each dataset, we generate $n = 20$ and $n = 1$ samples per problem with the LLM, respectively. We run the LLM-generated code completion against a predefined set of unit tests. For each unit test, we record the error type when running the generated program against that test case. We use the hyperparameters temperature $= 0.2$ and top $p = 0.95$. Table 4 presents our results for Python and Go. The columns standard and SynCode represent the total number of generated programs with syntax errors for the respective approaches. The column $\downarrow$ designates the percentage reduction in syntax errors from the standard generation to the SynCode generation. In this evaluation, across both HumanEval and MBXP datasets, we generate a total of 4154 samples for each language. On average, of all standard generated samples, 6% and 25% have syntax errors for Python and Go, respectively.

Notably, our experiments reveal that SynCode reduces the number of syntax errors by over 90% over the baseline in most experiments. Moreover, SynCode reduces the number of syntax errors to less than 1% of the total samples. Interestingly, we observe significantly more Syntax errors in standard LLM-generated Go code than in Python code, likely because the LLMs are trained more extensively on Python code than Go. Thus, SynCode can be especially effective for Go and more underrepresented programming languages, where LLMs are more likely to generate syntax errors due to a limited understanding of the language. SynCode can bridge this gap by guiding the LLM to sample only the syntactically valid tokens during decoding.

Table 5: Functional correctness on HumanEval problems

| Metric | Architecture | Python | | Go | |
|--------|-------------|--------|--------|--------|--------|
| | | Standard | SYNCODE | Standard | SYNCODE |
| pass@1 | CodeGen-350M | 6.8% ±0.1 | 6.9% ±0.1 | 3.6% ±0.0 | 3.6% ±0.0 |
| | WizardCoder-1B | 20.0% ±0.2 | 20.0% ±0.3 | 9.3% ±0.3 | 9.5% ±0.2 |
| | LLaMA-7B | 11.2% ±0.3 | 11.5% ±0.3 | 3.8% ±0.2 | 4.25% ±0.2 |
| pass@10 | CodeGen-350M | 10.4% ±0.3 | 10.6% ±0.3 | 5.6% ±0.2 | 6.1% ±0.3 |
| | WizardCoder-1B | 27.6% ±0.5 | 28.4% ±0.7 | 12.5% ±0.3 | 13.7% ±0.4 |
| | LLaMA-7B | 17.1% ±0.4 | 18.9% ±0.5 | 8.8% ±0.3 | 8.8% ±0.3 |

Table 6: DFA Mask store creation time and memory

| Model | $|V|$ | Python | | Go | |
|-------|-----|---------|--------|---------|--------|
| | | Time(s) | Memory | Time(s) | Memory |
| CodeGen-350M | 51200 | 602.26 | 1.87GB | 603.03 | 1.58GB |
| WizardCoder-1B | 49153 | 588.28 | 1.83GB | 588.84 | 1.54GB |
| LLaMA-7B | 32000 | 382.26 | 1.17GB | 380.49 | 1.06GB |

We further analyze the errors in Python and Go code generated by the LLMs augmented with SYNCODE, an example of which is presented in Appendix A.6. All of the errors were because the LLM failed to generate a complete program within the maximum token limit. Recall, SYNCODE provides guarantees of completeness for syntactically correct partial programs. However, it does not guarantee convergence to a syntactically correct and complete program.

**Functional Correctness for Code Generation.** We investigate whether augmenting LLMs with SYN-CODE improves the functional correctness of the generated code. We evaluate functional correctness using the pass@k metric, where $k$ samples are generated per problem, and a problem is considered solved if any sample passes a set of unit tests, and the fraction of solved problems is calculated. Table 5 reports our results for pass@1 and pass@10 for generated code completions to problems from the HumanEval dataset. We observe that augmenting LLMs with SYNCODE has a slight improvement in functional correctness over standard generation. This observation indicates that for these state-of-the-art models, syntactic correction can result in a small improvement in the logical correctness of the code.

## 6.4 Mask Store Overhead

We analyze the time and memory overhead involved in generating a DFA mask store using SYNCODE. The DFA mask store for Llama-2-7B-chat took 113.72 seconds to create and consumes 181 MB of memory. Additionally, we report the creation time and memory overhead of DFA mask stores for models used for Python and Go in Table 6. Each row shows the SYNCODE store generation time in seconds, and memory in GBs, for a particular LLM and grammar. The $|V|$ column represents the total vocabulary size of the tokenizer of the particular LLM. We see that generating the store requires less than 2GB of memory and several minutes across the evaluated models and grammars. This overhead is minimal for practical SYNCODE use cases, as the mask store is a one-time generation task. Thereafter, the mask store can be efficiently loaded into memory and used for repeated inference. We see smaller mask store generation time and memory with Llama-2-7B-chat and JSON grammar with 18 terminals as opposed to LLaMA-7B, WizardCoder-1B, and CodeGen-350M with Python and Go grammars with 94 and 87 terminals respectively since the size of the mask store is proportional to the number of terminals in the grammar.

## 7 Related Work

Our work focuses on enhancing the syntactical accuracy LLMs by using a constrained decoding algorithm. Prior research has explored two other primary directions to enhance LLMs' accuracy in generating formal language: 1) Fine-tuning or prompt engineering (Bassamzadeh and Methani, 2024; Weyssow et al., 2024), which demands substantial data, compute resources, and time, often without any formal guarantees. 2) Modifications to the LLM's architecture or tokenization (Murty et al., 2023; Dong et al., 2023; Zhu et al.,

Table 7: Overview of various constrained decoding methods

| | Regex | CFG | Precomputed | GPL | Max CFG | Input format |
|---|:---:|:---:|:---:|:---:|:---:|:---:|
| LMQL (Beurer-Kellner et al., 2023) | ✓ | ✗ | ✗ | ✗ | 50-100 | LMQL DSL |
| GUIDANCE (Lundberg et al., 2023) | ✓ | ✓ | ✗ | ✗ | 50-100 | Python DSL |
| OUTLINES (Willard and Louf, 2023) | ✓ | ✓ | ✓ | ✗ | 50-100 | Lark EBNF |
| PICARD (Scholak et al., 2021) | ✓ | ✓ | ✗ | ✗ | 50-100 | Haskell |
| SYNCHROMESH (Poesia et al., 2022) | ✓ | ✓ | ✗ | ✗ | ‡ | ANTLR |
| LLAMA.CPP (Gerganov and et. al., 2024) | ✓ | ✓ | ✗ | ✗ | 50-100 | GBNF DSL |
| GCD (Geng et al., 2023) | ✓ | ✓ | ✗ | ✗ | 50-100 | GF |
| DOMINO (Beurer-Kellner et al., 2024) | ✓ | ✓ | ✓ | ✗ | 50-100 | GBNF DSL |
| SYNCODE (ours) | ✓ | ✓ | ✓ | ✓ | 500+ | Lark EBNF |

† Implementation issues ‡ Synchromesh is closed-source and the information about DSL grammars is unavailable

GF: Grammatical Framework, GBNF is a DSL defined by LLAMA.CPP

2024), although these techniques have not yet achieved performance comparable to the current state-of-the-art standard LLMs. However, both fine-tuning and architectural changes are complementary to the grammar-guided decoding approach that we focus on in our work, and any gains through those techniques will improve the overall quality of LLM generation.

There are several recent works on constrained LLM generation (Wei et al., 2023; Beurer-Kellner et al., 2023; Lundberg et al., 2023; Willard and Louf, 2023; Scholak et al., 2021; Poesia et al., 2022; Gerganov and et. al., 2024; Geng et al., 2023; Beurer-Kellner et al., 2024; Agrawal et al., 2023; Melcer et al., 2024). This includes recent works that have used language-server (tools built for communication between IDEs and programming language-specific tools like static analyzers and compilers) suggestions to enforce language-specific semantic constraints during decoding (Agrawal et al., 2023; Wei et al., 2023). These techniques do not guarantee syntactical accuracy and rely on the availability and efficiency of language servers. More recent works such as (Ugare et al., 2025; Banerjee et al., 2025; Suresh et al., 2025) were built on top of SYNCODE.

**Structured LLM Generation.** We focus our further discussion on comparison to the techniques that constrain LLM for structured generation according to a formal language. We compare SYNCODE with prior works in terms of precision and efficiency of the algorithms and generality and scalability of frameworks. Table 7 presents the various recent techniques for structured LLM generation. The columns "Regex" and "CFG" indicate regular expression and CFG constraining features, respectively. The "Precomputed" column denotes techniques that precompute certain structures to enhance generation efficiency. The "GPL" column specifies if the tools support general-purpose PLs. "Max CFG" displays the number of production rules in the largest supported Grammar by these techniques. We obtained these numbers by examining the built-in grammars that were provided in the corresponding libraries. Finally, the "Input Format" column indicates the format used to specify generation constraints. In addition to the improvement over the baselines presented in the evaluation, our work focuses on rigorously formalizing the correctness of our CFG-guided generation approach.

Recent works such as GUIDANCE (Lundberg et al., 2023) and LMQL (Beurer-Kellner et al., 2023) mitigate the unpredictability of LLM responses by using template or constraint-based controlled generation techniques. These libraries feature a templating engine where prompts are expressed with holes for the generation to fill. LMQL (Beurer-Kellner et al., 2023) supports general regular expression constraints, but not CFG constraints. GUIDANCE (Lundberg et al., 2023) supports CFG-guided generation. It uses Earley parsing (Earley, 1970) for constrained decoding. Similar to other related works, it incurs high inference overhead as it checks the syntactical validity of the entire model vocabulary at each step. It uses a trie similar to (Poesia et al., 2022; Willard and Louf, 2023; Beurer-Kellner et al., 2024). As shown in our evaluation it incurs higher overhead for JSON generation than SYNCODE. It iterates over the vocabulary in order of the next token probability to efficiently compute the next token. However, this leads to a lack of generality and it cannot be directly combined with an arbitrary decoding strategy.

OUTLINES (Willard and Louf, 2023) is a library originally focused on regular expression-guided generation and recently extended to support grammar-guided generation. During LLM generation, OUTLINES employs

an incremental Lark-based LALR parser to determine the next acceptable terminals based on the grammar. Its approach differs from SynCode in how it processes acceptable tokens: Outlines computes an expensive union of regular expressions from all terminals during inference, converting this into a DFA, then validates tokens against this combined structure. In contrast, SynCode treats each sequence of acceptable terminals separately during decoding steps, avoiding the costly union operation. As shown in our evaluation (Section 6.1 and Section 6.2), SynCode performs better than Outlines on generating with JSON and SQL grammar and it currently lacks support for large GPL grammars.

llama.cpp (Gerganov and et. al., 2024), has also recently introduced support for grammar-guided generation. This approach models a nondeterministic pushdown automaton with $N$ stacks to maintain possible parse states. llama.cpp defines a new grammar syntax and implements a simplified basic parser in C++. While this implementation in C++ reduces some parsing overhead compared to heavier LR(1) parsers implemented in Python on top of Lark for SynCode, it is algorithmically inefficient. This inefficiency again is due to the requirement to iterate over the entire vocabulary and update stack states during inference. Moreover, the non-standard grammar syntax and limited support for general grammar features restrict its evaluation to simpler grammars such as JSON. We anticipate that llama.cpp and Outlines would perform even slower on grammars with more rules, terminals, and complex regular expressions, such as those found in Python and Go. As shown in our evaluation, SynCode is more efficient and results in fewer syntax errors.

Synchromesh (Poesia et al., 2022) is a proprietary a tool from Microsoft that supports CFG-guided syntactic decoding of LLMs. Similar to Outlines, it creates a union of regular expressions of terminals during LLM generation. Further, Synchromesh uses a non-incremental parser for parsing. Both of these lead to lower time complexity. Synchromesh uses techniques like Target Similarity Tuning for semantic example selection and Constrained Semantic Decoding to enforce user-defined semantic constraints and works on DSLs. In contrast, our work, SynCode focuses exclusively on syntactic generation.

Picard (Scholak et al., 2021) uses a specific decoding strategy that maintains a beam of multiple candidate outputs and promptly rejects the candidates that violate the syntax. It utilizes an incremental monadic parser and was developed specifically to support SQL generation. Introducing a new grammar into Picard necessitates considerable effort, as it lacks support for a grammar-defining language to provide grammar rules.

Recent work Domino (Beurer-Kellner et al., 2024) provides CFG-guided LLM generation, sharing conceptual similarities with SynCode while differing in key technical aspects. Domino avoids traversing the entire vocabulary during inference by precomputing a prefix tree corresponding to each NFA state of the grammar's terminals. This structure serves a purpose similar to SynCode's DFA mask store, but we believe our approach offers superior efficiency. SynCode's mask store leverages boolean mask operations that can be performed highly efficiently on modern hardware architectures, particularly GPUs, where parallelized bitwise operations provide significant performance advantages (Paszke et al., 2019b). This architectural difference contributes to SynCode's exceptional inference speed compared to alternatives. Domino defines the *minimally invasive* property, which is equivalent to SynCode's soundness property. However, a critical distinction between the two approaches lies in their approximation strategies: Domino applies an under-approximation approach, permitting only tokens that precisely align with the parser's lookahead, while SynCode adopts a conservative over-approximation approach, allowing tokens as long as their prefixes match the parser lookahead. This fundamental difference has important theoretical implications. Due to Domino's under-approximation strategy, it requires $\infty$ parser lookahead to achieve soundness guarantees, whereas SynCode ensures soundness for any lookahead depth. This gives SynCode a theoretical advantage in providing stronger correctness guarantees without the computational costs of extensive lookahead. Furthermore, SynCode demonstrates superior scalability in handling complex grammars. The largest grammar that Domino currently supports is a highly simplified C grammar with approximately 70 rules, incurring roughly 25% overhead. In contrast, SynCode efficiently handles substantially more complex grammars with lower computational overhead, as demonstrated in our experimental results. Unfortunately, Domino's implementation code is not publicly available yet, preventing a direct experimental comparison with SynCode. Nevertheless, our theoretical analysis and empirical results with other baselines strongly suggest that SynCode represents a significant advancement in CFG-guided LLM generation, combining stronger theoretical guarantees with practical efficiency.

**Fixed Schema Generation.** Many recent works perform constrained LLM decoding to ensure that the generated output follows a fixed schema of JSON or XML (Zheng et al., 2023; Beurer-Kellner et al., 2024; Willard and Louf, 2023; Sengottuvelu and et. al., 2024). When employing a fixed schema, many intermediate points in the generation process offer either a single syntactical choice (e.g., key in the JSON schema) or present only a handful of distinct options. In cases where only one choice exists, the generation of the next token through the LLM can be entirely skipped. Alternatively, when there are multiple but limited choices, techniques like speculative decoding can be used to expedite the generation process (Chen et al., 2023; Leviathan et al., 2023). SYNCODE does not focus on generation problems with fixed schema, it solely focuses on CFG-guided generation. We made the same observation as in (Beurer-Kellner et al., 2024), techniques such as speculation are not useful for CFGs where the schema is not fixed.

## 8 Conclusion

Existing methods for guiding LLMs to produce syntactically correct output have been notably slow and restrictive. In this paper, we present SYNCODE, an efficient and general framework to enhance LLMs' ability to generate syntactical output for various formal languages. During decoding, SYNCODE incrementally parses the partially generated output, computes the unparsed remainder and acceptable terminal sequences, and then leverages the remainder, accept sequences, and pre-computed DFA mask store to compute a mask to constrain the LLM's vocabulary to only syntactically valid tokens. We evaluated SYNCODE on generating syntactically correct JSON, SQL, Python, and Go code with different combinations of datasets, models, and tasks. SYNCODE eliminates syntax errors in JSON completions and significantly improves JSON schema validation over the baselines. Furthermore, SYNCODE reduces the number of syntax errors in generated Python and Go code by 96.07% on average compared to standard generation. We believe that our approach will pave the way for more efficient and higher-quality structured LLM generation in real-world applications.

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

# A   Appendix

## A.1   List of Symbols

| | |
|---|---|
| $G$ | Formal Grammar |
| $L(G)$ | Language of a grammar |
| $L_p(G)$ | Prefix language of a grammar |
| $l$ | lexical tokens |
| $l_i$ | $i$-th lexical token in the parsed output |
| $\tau$ | A terminal in the grammar |
| $\tau_i$ | Terminal type of $i$-th lexical token |
| $\Gamma$ | Set of all terminals in the grammar |
| $L^\Gamma(G)$ | Language of terminals for grammar $G$ |
| $L_p^\Gamma(G)$ | Prefix language of terminals |
| $P$ | Parser |
| $\Lambda$ | Sequence of terminals |
| $\mathcal{T}$ | Tokenizer in an LLM |
| $V$ | Vocabulary of an LLM |
| $V_k$ | Subset of vocabulary containing acceptable tokens at $k$-th LLM generation iteration |
| $\rho_\tau$ | Regular expression for a terminal $\tau$ |
| $\rho_i$ | Regular expression corresponding to $i$-th lexical token |
| $\preccurlyeq$ | Partial order over set of terminal sequences |
| $r$ | Remainder from SYNCODE parsing the partial output |
| $C_k$ | Partial output at $k$-th iteration of LLM generation |
| $C_k^\square$ | Parsed prefix of partial output $C_k$ at $k$-th iteration of LLM generation |
| $\mathcal{A}$ | Set of accept sequences |
| $\mathcal{M}_\alpha$ | DFA lookup store function for terminal sequences of length $\alpha$ |
| $dmatch$ | Match with DFA walk as defined in Section 4 |
| $pmatch$ | Partial match with regular expression |
| $pparse$ | Partial parsing function |
| $m$ | Boolean mask |
| $D$ | Deterministic finite automaton |
| $Q$ | States in a DFA |
| $\Sigma$ | Set of characters i.e. alphabet for DFA |
| $\delta$ | Transition function in a DFA |
| $\delta^*$ | Extended transition function in a DFA |
| $q_0$ | Start state of a DFA |
| $F$ | Set of final states in DFA |
| $live$ | Live states of the DFA |
| $Q_\Omega$ | Set containing all DFA states for DFAs of all terminals in the grammar |
| $A_0$ | Set of terminals acceptable for current lexical token |
| $A_1$ | Set of terminals acceptable as for next lexical token |
| $Lex$ | Lexer function |
| $len$ | Length of a sequence |
| $T_{cur}$ | Current set of tokens |
| $\mathcal{S}$ | Map for storing parser state |

## A.2 Proofs for Theorems

**Lemma 1.** *Given* $\Lambda = \{\tau_{f+1}, \tau_{f+2} \ldots \tau_{f+d}\}$, $\Lambda^p = \{\tau_{f+2} \ldots \tau_{f+d}\}$ *and* $\rho_\Lambda = (\rho_{f+1}, \rho_{f+2}, \ldots, \rho_{f+d})$, $dmatch(w, q_0^{\tau_1}, \Lambda^p) \iff pmatch(w, \rho_\Lambda)$.

*Proof.*    (a) First we prove $dmatch(w, q_0^{\tau_{f+1}}, \Lambda^p) \implies pmatch(w, \rho_\Lambda)$ We prove this using induction on the length $i$ of $w$.

For $i = 0$, $pmatch(w, \rho_\Lambda)$ is trivially true.

Now, we assume that for $w$ of length $i < k$, $dmatch(w, q_0^{\tau_{f+1}}, \Lambda^p) \implies pmatch(w, \rho_\Lambda)$.

We consider $w$ of length $k$ and $dmatch(w, q_0^{\tau_{f+1}}, \Lambda^p)$.

We consider 3 conditions from Definition 10.

If condition 1 is true, $\delta_{\tau_{f+1}}^*(w, q_0^{\tau_{f+1}}) \in live(Q_{\tau_{f+1}})$. Let $q_1 = \delta^*(w, q_0^{\tau_{f+1}})$. By Definition 9, $\exists w_1$ s.t. $\delta_{\tau_{f+1}}^*(w_1, q_1) \in F_{\tau_{f+1}}$. Hence,

$$\delta^*(w.w_1, q_0^{\tau_{f+1}}) \in F_{\tau_{f+1}} \implies w.w_1 \in L(\rho_{\tau_{f+1}})$$

We assume that each terminal $L(\tau_i)$ is non-empty. Hence,

$$\exists w_2 \in L(\rho_{\Lambda^p}) \implies w.w_1.w_2 \in L(\rho_\Lambda)$$

Hence, by condition 2 from Definition 8, $pmatch(w, \rho_\Lambda)$.

If condition 2 is true, $\exists w_1, w_2$ such that $w_1.w_2 = w$, $\delta_{\tau_{f+1}}^*(w_1, q_0^{\tau_{f+1}}) \in F$ and $\Lambda^p = \{\}$. Here, $w_1 \in L(\rho_{\tau_{f+1}})$. Since $\Lambda^p = \{\}$, $\rho_\Lambda = \rho_1$, and hence, $w_1 \in L(\rho_\Lambda)$. Hence by condition 1 from Definition 8, $pmatch(w, \rho_\Lambda)$.

If condition 3 is true, $\exists w_1, w_2$ such that $w_1.w_2 = w$, $\delta_{\tau_{f+1}}^*(w_1, q_0^{\tau_{f+1}}) \in F_{\tau_{f+1}}$, and $dmatch(w_2, q_0^{\tau_{f+2}}, \{\tau_{f+3} \ldots \tau_{f+d}\}) = true$.

$$\delta_{\tau_{f+1}}^*(w_1, q_0^{\tau_{f+1}}) \in F_{\tau_{f+1}} \implies w_1 \in L(\rho_{\tau_{f+1}})$$

Since length of $w_2 < k$, by our induction hypothesis, $pmatch(w_2, \rho_{\Lambda^p}) = true$. By Definition 8, there are two possibilities. Suppose $\exists w_2 = w_3.w_4$ such that $w_3 \in L(\rho_{\Lambda^p})$.

$$w_1.w_3 \in L(\rho_\Lambda) \implies pmatch(w, \rho_\Lambda) = true$$

Alternatively, if $\exists w_3$ such that $w_2.w_3 \in L(\rho_{\Lambda^p})$

$$w_1.w_2.w_3 \in L(\rho_\Lambda) \implies pmatch(w, \rho_\Lambda) = true$$

Hence, our induction proof is complete and $pmatch(w, \rho_\Lambda) = true$

(b) Next we prove $pmatch(w, \rho_\Lambda) \implies dmatch(w, q_0^{\tau_{f+1}}, \Lambda^p)$ We prove this using induction on the length $i$ of $w$.

For $i = 0$, $dmatch(w, q_0^{\tau_{f+1}}, \Lambda^p)$ is trivially true.

Now, we assume that for $w$ of length $i < k$, $pmatch(w, \rho_\Lambda) \implies dmatch(w, q_0^{\tau_{f+1}}, \Lambda^p)$

Now we consider $w$ of length $k$ and $pmatch(w, \rho_\Lambda)$.

By Definition 8, there are two possible conditions

**Case 1:** $\exists w_1 \in \Sigma^*, w_2 \in \Sigma^+$ such that $w = w_1.w_2$ and $w_1 \in L(\rho_\Lambda)$

Hence, $\exists w_3, w_4$ such that $w_1 = w_3.w_4$ and $w_3 \in L(\rho_{\tau_{f+1}})$ and $w_4 \in L(\rho_{\Lambda^p})$. By induction hypothesis,

$$pmatch(w_4.w_2, \rho_{\Lambda^p}) \implies dmatch(w_4 w_2, \{\tau_{f+2}, \tau_{f+3} \ldots \tau_{f+d}\})$$

Since $w = w_3.w_4.w_2$ and

$$w_3 \in L(\rho_{\tau_{f+1}}) \implies \delta_{\tau_{f+1}}^*(w_3, q_0^{\tau_{f+1}}) \in F_{\tau_{f+1}}$$

Hence, by condition 3 in Definition 10, $dmatch(w, q_0^{\tau_{f+1}}, \Lambda^p)$

**Case 2:** $\exists w_1$ such that $w.w_1 \in L(\rho_\Lambda)$
Hence, $\exists w_2, w_3$ s.t $w.w_1 = w_2.w_3$ and $w_2 \in L(\rho_{\tau_{f+1}})$ and $w_3 \in L(\rho_\Lambda)$
Now there are two possibilities, either $w$ is prefix of $w_2$ or $w_2$ is prefix of $w_2$
Supoose $w$ is prefix of $w_2$, then $\delta^*_{\tau_{f+1}}(w, q_0^{\tau_{f+1}}) \in live(Q_{\tau_{f+1}})$ and hence by Definition 10, $dmatch(w, q_0^{\tau_{f+1}}, \Lambda^p)$ Alternatively, if $w_2$ is prefix of $w$ then $\exists w_4$ s.t. $w = w_2 w_4$
Hence, $w_4.w_1 = w_3 \in L(\rho_{\tau_{f+1}})$ and thus $pmatch(w_4, \rho_{\Lambda^p})$
By induction hypothesis $dmatch(w_4, q_0^{\tau_{f+2}}, \{\tau_{f+3}, \tau_4 \dots \tau_{f+d}\})$
and since $w = w_2.w_4$ and $\delta^*_{\tau_{f+1}}(w_2, q_0^{\tau_{f+1}}) \in F_{\tau_{f+1}}$. We get $dmatch(w, q_0^{\tau_{f+1}}, \Lambda^p)$

$\square$

**Lemma 2.** *If $q = \delta^*_\tau(r, q_0^\tau)$ and no prefix of $r$ is in $L(\tau)$ i.e. $\nexists w_1 \in \Sigma^*, w_2 \in \Sigma^*$ such that $w_1.w_2 = r$ and $\delta^*_\tau(w_1, q_0^\tau) \in F_\tau$ then $dmatch(t, q, \Lambda) \iff dmatch(r.t, q_0^\tau, \Lambda)$.*

*Proof.*     (a) First, we prove $dmatch(t, q, \Lambda) \implies dmatch(r.t, q_0^\tau, \Lambda)$.
From Definition 10, either of the 3 conditions hold true for $dmatch(t, q, \Lambda)$.

If condition 1 is true then

$$\delta^*_{\tau_1}(t, q) \in live(Q_\tau) \implies \delta^*_\tau(r.t, q_0^\tau) \in live(Q_\tau) \implies dmatch(r.t, q_0^\tau, \Lambda)$$

If condition 2 is true, $\exists w_1, w_2$ such that $w_1.w_2 = t$, $\delta^*_\tau(w_1, q) \in F$ and $\Lambda = \{\}$. Therefore,

$$\delta^*_\tau(r.w_1, q) \in F \implies dmatch(r.t, q_0^\tau, \Lambda)$$

If condition 3 is true, $\exists w_1, w_2$ such that $w_1.w_2 = t$, $\delta^*_\tau(w_1, q) \in F$
and $dmatch(w_2, q_0^{\tau_1}, \{\tau_2 \dots \tau_d\}) = true$. Therefore,

$$\delta^*_\tau(r.w_1, q) \in F \implies dmatch(r.t, q_0^\tau, \Lambda)$$

Therefore, in all cases, $dmatch(rt, q_0^\tau, \Lambda)$ must hold.

(b) Now, we prove $dmatch(rt, q_0^\tau, \Lambda) \implies dmatch(t, q, \Lambda)$.

From Definition 10, either of the 3 conditions hold true for $dmatch(r.t, q_0^\tau, \Lambda)$.

If condition 1 is true then

$$\delta^*_{\tau_1}(r.t, q_0^\tau) \in live(Q_\tau) \implies \delta^*_\tau(t, q) \in live(Q_\tau) \implies dmatch(t, q, \Lambda)$$

If condition 2 is true, $\exists w_1, w_2$ such that $w_1.w_2 = r.t$, $\delta^*_\tau(w_1, q_0^\tau) \in F$ and $\Lambda = \{\}$. Since no prefix of $r$ is accepted by $L(\tau)$, $\exists w_3$ s.t. $w_3 w_4 = t$ and

$$\delta^*_\tau(w_3, q) \in F \implies dmatch(t, q, \Lambda)$$

If condition 3 is true, $\exists w_1, w_2$ such that $w_1.w_2 = r.t$, $\delta^*_\tau(w_1, q_0^\tau) \in F$
and $dmatch(w_2, q_0^{\tau_1}, \{\tau_2 \dots \tau_d\}) = true$. Since no prefix of $r$ is accepted by $L(\tau)$, $\exists w_3$ s.t. $w_3 w_4 = t$ and

$$\delta^*_\tau(w_3, q) \in F \implies dmatch(t, q, \Lambda)$$

Therefore, in all cases, $dmatch(t, q, \Lambda)$ must hold.

$\square$

**Theorem 1.** *Let $C_k \in L_p(G)$ be the partial output and any integer $d \geq 1$, let $\mathcal{A}_d \subseteq \Gamma^d$ contain all possible accept terminal sequences of length $d$ and $r \in \Sigma^*$ denote the remainder. If $m = GrammarMask(\mathcal{A}, r)$ then for any $t \in V$, if $C_k.t \in L_p(G)$ then $t \in set(m)$*

*Proof.* Let $r, \Lambda^{\square} = pparse(C_k)$ where $\Lambda^{\square} = \tau_1, \tau_2 \ldots \tau_f$ and let $r_1, \Lambda_1 = pparse(C_k.t)$ where $\Lambda_1 = \tau_1, \tau_2 \ldots \tau_f \ldots \tau_{f+g}$

Hence, we can split $r.t$ such that for $w \in \Sigma^*$, $r.t = w.r_1$ and $w \in L(\tau_{f+1} \ldots \tau_{f+g})$

There are two possible cases:

**Case 1:** $g < d$

$$w \in L(\tau_{f+1} \ldots \tau_{f+g})$$

$$\implies w \in L_p(\tau_{f+1} \ldots \tau_{f+g})$$

By our assumption on $\mathcal{A}_d$ there must exist $\Lambda_2 = \tau_{f+1} \ldots \tau_{f+d}$ s.t. $\tau_{f+1} \ldots \tau_{f+g}$ is prefix of $\Lambda_2$. Hence,

$$\implies w \in L_p(\Lambda_2)$$

$$\implies pmatch(r.t, \Lambda_2)$$

**Case 2:** $g \geq d$

Since we assume that $\mathcal{A}_d$ contains all possible accept sequence of length $d$, $\Lambda_2 = \tau_{f+1} \ldots \tau_{f+d}$ must be contained in $\mathcal{A}_d$

Hence, $\exists w_1, w_2 \in \Sigma^*$ such that $w = w_1.w_2$ and

$$w_1 \in L(\Lambda_2)$$

$$\implies w \in L_p(\Lambda_2)$$

$$\implies pmatch(r.t, \Lambda_2)$$

In both cases, $pmatch(r.t, \Lambda_2)$. Using Lemma 1,

$$\implies dmatch(r.t, q_0^{\tau_{f+1}}, \{\tau_{f+2} \ldots \tau_{f+d}\})$$

Using Lemma 2 if $q = \delta_{\tau_{f+1}}^*(r, q_0^{\tau_{f+1}})$

$$dmatch(r.t, q_0^{\tau_{f+1}}, \{\tau_{f+2} \ldots \tau_{f+d}\}) \implies dmatch(t, q, \{\tau_{f+2} \ldots \tau_{f+d}\})$$

Here from Definition 12, if $\mathcal{M}_{d-1}(q, \{\tau_{f+2} \ldots \tau_{f+d}\}) = m_2$ then $t \in set(m_2)$.

Since $m_2 \subseteq m$, we have our result $t \in set(m)$.

$\square$

**Lemma 3.** *Given $\mathcal{A}_1$ and $\mathcal{A}_2$ are set of accept sequences such that $\mathcal{A}_1 \preccurlyeq \mathcal{A}_2$ and $m_1 = GrammarMask(\mathcal{A}_1, r)$ and $m_2 = GrammarMask(\mathcal{A}_2, r)$ then $set(m_2) \subseteq set(m_1)$*

*Proof.* Since $\forall \Lambda_2 \in \mathcal{A}_2 \exists \Lambda_1 \in \mathcal{A}_1 \exists \Lambda_3 \in \Gamma^*$ s.t. $\Lambda_2 = \Lambda_1.\Lambda_3$, Hence

$$pmatch(w, \rho_{\Lambda_2}) \implies pmatch(w, \rho_{\Lambda_1})$$

Hence, for the mask $set(m_2) \subseteq set(m_1)$

$\square$

**Theorem 2.** *Let $C_k \in L_p(G)$ be the partial output, let $\mathcal{A}_d \subseteq \Gamma^d$ contain all possible accept terminal sequences of length $d$ and $r \in \Sigma^*$ denote the remainder. Suppose for any $t \in V, d > len(t)$ and $m = GrammarMask(\mathcal{A}_d, r)$ such that $t \in set(m)$ then $C_k.t \in L_p(G)$*

For the simplicity of presenting the proof, we assume that $d > 2$.

Since $t \in set(m)$ for some $\Lambda_1 = \{\tau_{f+1}, \tau_{f+2} \ldots \tau_{f+d}\} \in \mathcal{A}$

$$\implies dmatch(t, q, \{\tau_{f+2} \ldots \tau_{f+d}\}) \implies dmatch(r.t, q_0^{\tau_{f+1}}, \{\tau_{f+2} \ldots \tau_{f+d}\})$$

$$\implies pmatch(r.t, \{\rho_{\tau_{f+1}}.\rho_{\tau_{f+2}} \ldots \rho_{\tau_{f+d}}\})$$

By Definition 8, there are two possible cases:

1. $\exists w_1 \in \Sigma^*, w_2 \in \Sigma^+$ such that $r.t = w_1.w_2$ and $w_1 \in L(\rho_{\tau_{f+1}}.\rho_{\tau_{f+2}} \cdots \rho_{\tau_{f+d}})$
   We show that this case is not possible since our terminal sequence $\Lambda_1$ is long enough that no prefix of $r.t$ cannot be in $L(\rho_{\tau_{f+1}}.\rho_{\tau_{f+2}} \cdots \rho_{\tau_{f+d}})$
   We can infer that $len(w_1) < len(r.t) \implies len(w_1) < len(r) + len(t)$
   Further, from the assumption $d > len(t)$, we have

$$len(w_1) < d + len(r)$$

   Firstly, note that $r \notin L(\rho_{\tau_{f+1}}.\rho_{\tau_{f+2}})$ by the definition of remainder $r$
   Note that we assume no terminal contains empty string i.e. $\epsilon \notin L(\rho_{\tau_i})$
   Hence, every string in $L(\rho_{\tau_{f+2}} \ldots \rho_{\tau_{f+d}})$ should have length at least $d - 1$

   Clearly, $r$ is prefix of $w_1$. Let $w_3 \in \Sigma^*$, $r.w_3 = w_1$ and hence $len(w_3) > d - 1$
   Hence,

$$len(r) + d - 1 < len(w_1)$$

$$len(r) + d - 1 < len(w_1) < d + len(r)$$

   This is not possible and hence such $w_1$ cannot exist.

2. $\exists w_1 \in \Sigma^*$ such that $r.t.w_1 \in L(\rho_{\tau_{f+1}}.\rho_{\tau_{f+2}} \cdots \rho_{\tau_{f+d}})$

   By Definition 5, we have $\Lambda^\square, r = pparse(C_k)$ s.t $C_k = C_k^\square.r$, $\Lambda^\square = \tau_1, \tau_2 \ldots \tau_f$ $C_k^\square \in L(\rho_{\tau_1}.\rho_{\tau_2} \ldots \rho_{\tau_f})$.
   Let $\Lambda_1 = \tau_{f+1}, \tau_{f+2} \ldots \tau_{f+d}$
   Since, $C_k.t = C_k^\square.r.t$, $C_k^\square \in L(\Lambda^\square)$ and $r.t.w_1 \in L(\Lambda_1)$, we have

$$C_k^\square.r.t.w_1 \in L(\Lambda^\square.\Lambda_1)$$

$$C_k.t.w_1 \in L(\Lambda^\square.\Lambda_1)$$

   By Definition 7 of accept sequence, $\Lambda^\square.\Lambda_1 \in L_p^\Gamma(G)$, Hence

$$C_k.t.w_1 \in L_p(G) \implies C_k.t \in L_p(G)$$

Thus, our proof is complete and $C_k.t \in L_p(G)$

### A.3 Incremental Parsing Algorithm

Our parsing algorithm achieves incrementality in LLM generation by utilizing a map $\mathcal{S}$ to store the parser state. This map associates a list of lexical tokens with the corresponding parser state after parsing those tokens. Frequently, in subsequent LLM generation iterations, the count of lexical tokens remains the same—either the next vocabulary token is appended to the final lexical token, or it increases. Although uncommon, there are cases where the number of parsed lexical tokens may decrease during iterations. For example, in Python, an empty pair of double quotes, `""`, is recognized as a complete lexical token representing an empty string. On the other hand, `"""` serves as a prefix to a docstring, constituting an incomplete parser token. Consequently, the addition of a single double quote `"` reduces the overall count of lexical tokens in these iterations. We observe that while the total count of lexer tokens at the end may undergo slight changes during these iterations, the majority of prefixes of the parsed lexical tokens remain consistent. Hence, we establish a mapping between lists of prefixes of lexical tokens and the corresponding parser state after parsing those tokens. Subsequently, when parsing a new list of lexer tokens, we efficiently determine the maximum length prefix of the lexer token list that is already present in $\mathcal{S}$. This incremental approach significantly reduces the complexity of our parsing algorithm.

---

**Algorithm 4** Incremental Parsing

**Inputs:** $C_k$: partial output, $\mathcal{S}$: state map

1: **function** PARSE($C_k$)
2: $\quad l_1, l_2 \ldots l_f \leftarrow Lex(C_k)$
3: $\quad \gamma, S_\gamma \leftarrow \text{RestoreState}(\mathcal{S}, L)$
4: $\quad P \leftarrow \text{Initialize}(S_\gamma)$
5: $\quad parsed \leftarrow l_1.l_2 \ldots l_{\gamma-1}$
6: $\quad$ **for** $l_i \in l_\gamma, l_{\gamma+1} \ldots l_f$ **do**
7: $\quad\quad Next(P, l_i)$
8: $\quad\quad$ **if** $P.state = Error$ **then**
9: $\quad\quad\quad$ break
10: $\quad\quad parsed \leftarrow parsed + l_i$
11: $\quad\quad A_0 \leftarrow A_1$
12: $\quad\quad A_1 \leftarrow Follow(P)$
13: $\quad\quad S_i \leftarrow P.state$
14: $\quad\quad \text{Store}(\mathcal{S}, parsed, S_i)$
15: $\quad$ **if** $C_k = parsed$ **then**
16: $\quad\quad r = l_f$
17: $\quad\quad \mathcal{A} \leftarrow \{\tau_f, A_1[0]\}, \{\tau_f, A_1[1]\} \ldots \}$
18: $\quad\quad\quad \cup \{A_0[0]\}, \{A_0[1]\} \ldots \}$
19: $\quad$ **else**
20: $\quad\quad r = C_k - parsed$
21: $\quad\quad \mathcal{A} \leftarrow \{A_1[0]\}, \{A_1[1]\} \ldots$
22: $\quad$ **return** $\mathcal{A}, r$

---

While it could be feasible to introduce incrementality in the lexing operation, our experiments revealed that lexing consumes insignificant time in comparison to parsing. As a result, we opted to focus only on performing parsing incrementally.

Our incremental parsing algorithm uses a standard non-incremental base parser $P$ that maintains a parser state and supports two functions *Next* and *Follow*. The *Next* function accepts the next lexer token and then updates the parser state. The *Follow* function returns a list of acceptable terminals at the current parser state. These functions are present in common parser generator tools (Lark, ; ANTLR, ).

The Algorithm 4 presents our incremental parsing algorithm. The algorithm utilizes a lexer to tokenize the partial output. The function RestoreState is used to restore the state of the parser to the maximal matching prefix of lexical tokens that exist in $\mathcal{S}$. The main loop iterates through the tokens and maintains a parser state map. For each token, it updates the parser state, stores the mapping in $\mathcal{S}$, and retrieves the next set of acceptable terminals. The process continues until the end of the partial output. The algorithm returns accept sequences $\mathcal{A}$ and remainder $r$.

### A.4 Ablation Studies

In this section, we perform an ablation study for incremental parsing and max new tokens.

**Incremental Parsing.** We compare the runtime efficiency of utilizing incremental parsing over re-running parsing from scratch in SYNCODE. We run inference on CodeGen-350M with SYNCODE using incremental parsing and parsing from scratch on Python prompts from the HumanEval dataset. We generate $n = 1$ samples and control the max new tokens in the code completion. Our results are presented in Figure 10b, where the x-axis represents the max new tokens and the y-axis represents the average generation time, in

Table 8: SynCode on few-shot prompting

| Architecture | Error Type | Standard | SynCode | ↓ |
|---|---|---|---|---|
| CodeGen-350M | Syntax | 53 | 0 | 100% |
| | Indentation | 15 | 3 | 80% |
| WizardCoder-1B | Syntax | 40 | 2 | 95% |
| | Indentation | 22 | 1 | 95% |
| Llama-7B | Syntax | 110 | 0 | 100% |
| | Indentation | 40 | 5 | 88% |

seconds, with and without incremental parsing. As shown in the figure, the average generation time when re-parsing from scratch increases significantly as the maximum length of code that the LLM can generate increases. On the other hand, the average generation time increases slowly with incremental parsing. For max new tokens = 300, SynCode with incremental parsing achieves 9x speedup over running parsing from scratch. Our results collectively demonstrate that augmenting SynCode with incremental parsing dramatically improves generation time, especially when generating longer completions.

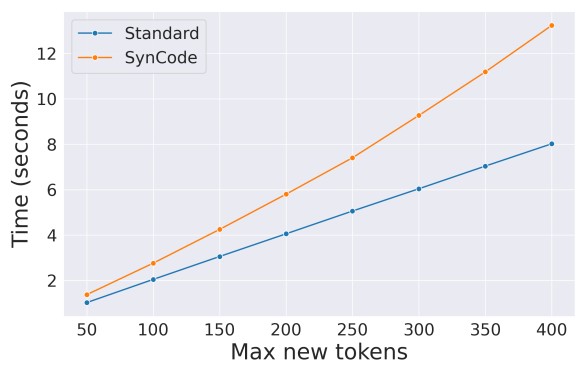

(a) Average generation time for different max new tokens

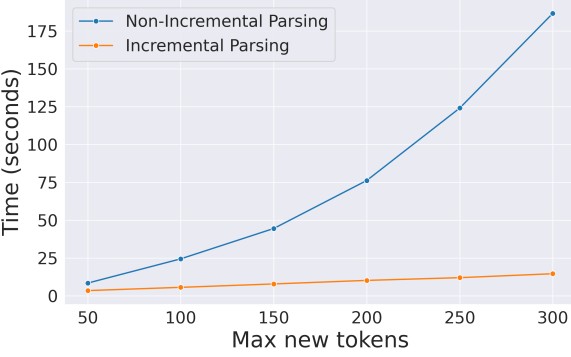

(b) Average generation time with and without incremental parser

Figure 10: Ablation studies on CodeGen-350M model.

**Max New Tokens.** We conduct an ablation study into the relationship between the maximum length of code that the LLMs can generate and generation times. We used Python prompts from the HumanEval dataset and leveraged CodeGen-350M to generate the code completions, both with and without the augmentation of the SynCode. As shown in Figure 10a, as we increase the max new tokens, we observe a corresponding increase in generation time.

### A.5 Few-Shot Prompting

Few-shot prompting (Ren et al., 2018) refers to the idea that language models do not need to be specifically trained for a downstream task such as classification or question answering. Rather, it is sufficient to train them on broad text-sequence prediction datasets and to provide context in the form of examples when invoking them. We study the performance of utilizing SynCode on few-shot prompting code generation tasks. We selected Python few-shot examples from the MBXP dataset and generated code completions with CodeGen-350M, LLaMA-7B, and WizardCoder-1B with SynCode and the standard no-masking generation. We present our results in Table 8. The columns standard and SynCode represent the total number of errors of a particular Error Type of LLM-generated code completions to problems in a particular dataset when utilizing that respective generation approach. The column ↓ represents the percentage reduction from the standard column to the SynCode column. As shown in the table, SynCode exhibits effectiveness not only in zero-shot but also in the context of few-shot prompting tasks. This signifies the versatility of SynCode in enhancing code generation across different prompt engineering techniques.

```python
def max_fill(grid, capacity):
    """You are given a rectangular grid of wells. Each row represents a single well,
       and each 1 in a row represents a single unit of water.
       Each well has a corresponding bucket that can be used to extract water from it,
       and all buckets have the same capacity. Your task is to use the buckets to empty the wells.
       Output the number of times you need to lower the buckets."""
    if len(grid) < 2:
        return 0
    if len(grid) == 1:
        return 1
    if len(grid) == 2:
        return grid[0][1] - grid[0][0]
    if len(grid) == 3:
        return grid[0][1] - grid[0][0] - grid[1][1]
    … 11 more lines
    if len(grid) == 9:
        return grid[0][1] - grid[0][0] - grid[1][1] - grid[2][1] - grid[
```

Figure 11: Syntactically Incorrect SYNCODE Program

## A.6 SynCode Syntax Errors

Figure 11 presents an example of when the SYNCODE augmented LLM fails to generate a complete program within the maximum token limit for a problem from the HumanEval dataset. While the code is a syntactically correct partial program, it is not a syntactically correct complete program. Recall, that SYNCODE guarantees completeness for syntactically correct partial programs but does not guarantee termination with a syntactically correct complete program.

### A.7 Prompts Used in the Evaluation

```
1  [INST] <<SYS>>
2  You are a helpful assistant that answers in JSON. Here's the json schema you must adhere to:
3  <schema>
4  {'title': 'Person', 'type': 'object', 'properties': {'firstName': {'type': 'string', '
       description': "The person's first name."}, 'lastName': {'type': 'string', 'description':
        "The person's last name."}, 'age': {'description': 'Age in years which must be equal to
        or greater than zero.', 'type': 'integer', 'minimum': 0}}, 'required': ['firstName', '
       lastName', 'age']}
5  </schema>
6
7  <</SYS>>
8
9  Please generate a JSON output for a person's profile that includes their first name, last
       name, and age. The first name should be 'Alice', the last name 'Johnson', and the age
       35. [/INST]
```

Listing 1: Example original JSON Prompt from the JSON-Mode-Eval dataset (NousResearch, 2024). The prompt consists of a system message that specifies a schema and a user message requesting JSON output given certain parameters.

```
1  [INST] <<SYS>>
2  You are a helpful assistant that answers in JSON. Here's the json schema you must adhere to:
3  <schema>
4  {'title': 'Person', 'type': 'object', 'properties': {'firstName': {'type': 'string', '
       description': "The person's first name."}, 'lastName': {'type': 'string', 'description':
        "The person's last name."}, 'age': {'description': 'Age in years which must be equal to
        or greater than zero.', 'type': 'integer', 'minimum': 0}}, 'required': ['firstName', '
       lastName', 'age']}
5  </schema>
6
7  <</SYS>>
8
9  Please generate a JSON output for a person's profile that includes their first name, last
       name, and age. The first name should be 'Alice', the last name 'Johnson', and the age
       35. Output only JSON. [/INST]
```

Listing 2: Example JSON prompt from the JSON-Mode-Eval dataset (NousResearch, 2024) after augmentation with an explicit request to only output JSON.

```
1
2  db_id: concert_singer
3  db_info: # stadium ( stadium_id , location , name , capacity , highest , lowest , average )
4  # singer ( singer_id , name , country , song_name , song_release_year , age , is_male )
5  # concert ( concert_id , concert_name , theme , stadium_id , year )
6  # singer_in_concert ( concert_id , singer_id )
7  # concert.stadium_id = stadium.stadium_id
8  # singer_in_concert.singer_id = singer.singer_id
9  # singer_in_concert.concert_id = concert.concert_id
10
11 question: How many singers do we have? Only output the SQL query.
12 SQL:
```

Listing 3: text-2-SQL prompt.

```
1  def has_close_elements(numbers: List[float], threshold: float) -> bool:
2      """ Check if in given list of numbers, are any two numbers closer to each other than
3      given threshold.
4      >>> has_close_elements([1.0, 2.0, 3.0], 0.5)
5      False
6      >>> has_close_elements([1.0, 2.8, 3.0, 4.0, 5.0, 2.0], 0.3)
7      True
8      """
```

Listing 4: Example Python prompt from the HumanEval dataset (Athiwaratkun et al., 2023)

```
1  package main
2
3  import (
4    "encoding/json"
5    "reflect"
6  )
7  // You're an expert Golang programmer
8  // Check if in given list of numbers, are any two numbers closer to each other than
```

```
 9  // given threshold.
10  // >>> has_close_elements([1.0, 2.0, 3.0], 0.5)
11  // False
12  // >>> has_close_elements([1.0, 2.8, 3.0, 4.0, 5.0, 2.0], 0.3)
13  // True
14  //
15  func has_close_elements (numbers []float64, threshold float64) bool {
```

Listing 5: Example Go prompt from the HumanEval dataset  (Athiwaratkun et al., 2023)

## A.8   Grammars Used in the Evaluation

### A.8.1   JSON Grammar

```
 1  ?start: value
 2
 3  ?value: object
 4  | array
 5  | UNESCAPED_STRING
 6  | SIGNED_NUMBER      -> number
 7  | "true"             -> true
 8  | "false"            -> false
 9  | "null"             -> null
10
11  array  : "[" [value ("," value)*] "]"
12  object : "{" [pair ("," pair)*] "}"
13  pair   : UNESCAPED_STRING ":" value
14
15  UNESCAPED_STRING: /\"[^"]*\"/
16
17  DIGIT: "0".."9"
18  HEXDIGIT: "a".."f"|"A".."F"|DIGIT
19  INT: DIGIT+
20  SIGNED_INT: ["+"|"-"] INT
21  DECIMAL: INT "." INT? | "." INT
22
23
24  _EXP: ("e"|"E") SIGNED_INT
25  FLOAT: INT _EXP | DECIMAL _EXP?
26  NUMBER: FLOAT | INT
27  SIGNED_NUMBER: ["+"|"-"] NUMBER
28  WS: /[ \t\f\r\n]/+
29
30  %ignore WS
```

Listing 6: JSON Grammar

### A.8.2   SQL Grammar

```
 1
 2  start: set_expr ";"? -> final
 3
 4  set_expr: query_expr
 5          | set_expr "UNION"i ["DISTINCT"i] set_expr -> union_distinct
 6          | set_expr "UNION"i "ALL"i set_expr -> union_all
 7          | set_expr "INTERSECT"i ["DISTINCT"i] set_expr -> intersect_distinct
 8          | set_expr "EXCEPT"i ["DISTINCT"i] set_expr -> except_distinct
 9          | set_expr "EXCEPT"i "ALL"i set_expr -> except_all
10
11  query_expr: select [ "ORDER"i "BY"i (order_by_expr ",")*  order_by_expr] [ "LIMIT"i
        limit_count [ "OFFSET"i skip_rows ] ]
12
13  select: "SELECT"i [SELECT_CONSTRAINT] [(select_expr ",")*] select_expr "FROM"i [(from_expr
        ",")*] from_expr [ "WHERE"i where_expr ] [ "GROUP"i "BY"i [(groupby_expr ",")*]
        groupby_expr ] [ "HAVING"i having_expr] [ "WINDOW"i window_expr ]
14
15  where_expr: bool_expression
16
17  select_expr.0: expression_math [ "AS"i alias ] -> select_expression
18
19  ?from_expr: from_item -> from_expression
20
21  order_by_expr: order -> order_by_expression
22
```

```
23  having_expr: bool_expression
24
25  groupby_expr: expression -> group_by
26
27  window_expr: [window_expr ","] _window_name "AS"i ( window_definition )
28
29  from_item: table_name [ "AS"i alias ] -> table
30             | join -> join
31             | cross_join -> cross_join_expression
32             | subquery
33  table_name: name
34
35  subquery: ( "(" (query_expr | join | cross_join) ")" ) [ "AS"i alias ]
36
37  cross_join: from_item "CROSS"i "JOIN"i from_item
38  join: from_item JOIN_EXPR from_item [ "ON"i bool_expression ] -> join_expression
39
40  JOIN_EXPR.5: (JOIN_TYPE WS)? "JOIN"i
41  JOIN_TYPE: "INNER"i | "OUTER"i? | JOIN_DIRECTION (WS "OUTER"i)? | JOIN_DIRECTION
42  JOIN_DIRECTION: "FULL"i | "LEFT"i | "RIGHT"i
43
44  ?expression_math: expression_product
45                  | expression_math "+" expression_product -> expression_add
46                  | expression_math "-" expression_product -> expression_sub
47                  | "CASE"i (when_then)+ "ELSE"i expression_math "END"i -> case_expression
48                  | "CAST"i "(" expression_math "AS"i TYPENAME ")" -> as_type
49                  | "CAST"i "(" literal "AS"i TYPENAME ")" -> literal_cast
50                  | AGGREGATION expression_math ")" [window_form] -> sql_aggregation
51                  | "RANK"i "(" ")" window_form -> rank_expression
52                  | "DENSE_RANK"i "(" ")" window_form -> dense_rank_expression
53                  | "COALESCE"i "(" [(expression_math ",")*] expression_math ")" ->
                        coalesce_expression
54                  | subquery -> subquery_expression
55
56  window_form: "OVER"i "(" ["PARTITION"i "BY"i (partition_by ",")* partition_by] ["ORDER"i "BY
        "i (order ",")* order [ row_range_clause ] ] ")"
57
58  partition_by: expression_math
59
60  row_range_clause: ( ROWS | RANGE ) frame_extent
61  frame_extent: frame_between | frame_preceding
62  frame_between: "BETWEEN"i frame_bound "AND"i frame_bound
63  frame_bound: frame_preceding | frame_following | "CURRENT"i "ROW"i
64  frame_preceding: UNBOUNDED PRECEDING | INT_NUMBER PRECEDING
65  frame_following: UNBOUNDED FOLLOWING | INT_NUMBER FOLLOWING
66  RANGE: "RANGE"i
67  ROWS: "ROWS"i
68  UNBOUNDED: "UNBOUNDED"i
69  PRECEDING: "PRECEDING"i
70  FOLLOWING: "FOLLOWING"i
71
72  when_then: "WHEN"i bool_expression "THEN"i expression_math
73  order: expression_math ["ASC"i] -> order_asc
74         | expression_math "DESC"i -> order_desc
75
76
77  ?expression_product: expression_parens
78                  | expression_product "*" expression_parens -> expression_mul
79                  | expression_product "/" expression_parens -> expression_div
80
81  ?expression_parens: expression
82                  | "(" expression_parens "*" expression ")" -> expression_mul
83                  | "(" expression_parens "/" expression ")" -> expression_div
84                  | "(" expression_parens "+" expression ")" -> expression_add
85                  | "(" expression_parens "-" expression ")" -> expression_sub
86
87  column_name: [name "."] (name | STAR)
88  ?expression: column_name -> column_name
89             | literal
90
91
92  SELECT_CONSTRAINT.9: "ALL"i | "DISTINCT"i
93  TYPENAME:  "object"i
94          | "varchar"i
95          | "integer"i
96          | "int16"i
97          | "smallint"i
98          | "int32"i
99          | "int64"i
100         | "int"i
101         | "bigint"i
```

```
102              | "float16"i
103              | "float32"i
104              | "float64"i
105              | "float"i
106              | "bool"i
107              | "datetime64"i
108              | "timestamp"i
109              | "time"i
110              | "date"i
111              | "cateSQLry"i
112              | "string"i
113   AGGREGATION.8: ("SUM("i | "AVG("i | "MIN("i | "MAX("i | "COUNT("i "DISTINCT"i | "COUNT("i)
114   alias: name -> alias_string
115   _window_name: name
116   limit_count: INT_NUMBER -> limit_count
117   skip_rows: INT_NUMBER
118   bool_expression: bool_parentheses
119                    | bool_expression "AND"i bool_parentheses -> bool_and
120                    | bool_expression "OR"i bool_parentheses -> bool_or
121   bool_parentheses: comparison_type
122                    | "(" bool_expression "AND"i comparison_type ")" -> bool_and
123                    | "(" bool_expression "OR"i comparison_type ")" -> bool_or
124                    | "EXISTS"i subquery -> exists
125   comparison_type: equals | not_equals | greater_than | less_than | greater_than_or_equal
126   | less_than_or_equal | between | in_expr | not_in_expr | subquery_in | subquery_not_in |
         is_null | is_not_null | like_expr | not_like_expr
127
128   equals: expression_math "=" expression_math
129   is_null: expression_math "IS"i "NULL"i
130   is_not_null: expression_math "IS"i "NOT"i "NULL"i
131   not_equals: expression_math ("<>" | "!=") expression_math
132   greater_than: expression_math ">" expression_math
133   less_than: expression_math "<" expression_math
134   greater_than_or_equal: expression_math ">=" expression_math
135   less_than_or_equal: expression_math "<=" expression_math
136   between: expression_math "BETWEEN"i expression_math "AND"i expression_math
137
138   // `LIKE` and `NOT LIKE`
139   like_expr: expression_math "LIKE"i expression_math
140   not_like_expr: expression_math "NOT"i "LIKE"i expression_math
141
142   // `IN` and `NOT IN`
143   in_expr: expression_math "IN"i "(" [expression_math ","]* expression_math ")"
144   subquery_in: expression_math "IN"i subquery
145   not_in_expr: expression_math "NOT"i "IN"i "(" [expression_math ","]* expression_math ")"
146   subquery_not_in: expression_math "NOT"i "IN"i subquery
147
148   ?literal: boolean -> bool
149           | number_expr -> number
150           | /'([^'])+'|''/ -> string
151           | timestamp_expression -> timestamp_expression
152   boolean: "TRUE"i -> true
153          | "FALSE"i -> false
154   ?number_expr: product
155
156   ?product: INT_NUMBER -> integer
157           | FLOAT -> float
158
159   INT_NUMBER: /[1-9][0-9]*/
160
161   STAR: "*"
162   window_definition:
163   timestamp_expression: "NOW"i "(" ")" -> datetime_now
164                        | "TODAY"i "(" ")" -> date_today
165
166   date: YEAR "-" MONTH "-" DAY
167   YEAR: /[0-9]{4}/
168   MONTH: /[0-9]{2}/
169   DAY: /[0-9]{2}/
170   time: HOURS ":" MINUTES ":" SECONDS
171   HOURS: /[0-9]{2}/
172   MINUTES: /[0-9]{2}/
173   SECONDS: /[0-9]{2}/
174   name: CNAME | ESCAPED_STRING
175
176   _STRING_INNER: /(?:[^"\\]|\\.)*?/
177   ESCAPED_STRING: "\"" _STRING_INNER "\""
178
179   %import common.CNAME
180   %import common.WS
181   %import common.SQL_COMMENT
```

```
182  %import common.WS_INLINE
183  %import common.FLOAT
184
185  %ignore WS
186  %ignore SQL_COMMENT
```

Listing 7: SQL Grammar

### A.8.3  Python Grammar

```
1   single_input: _NL | simple_stmt | compound_stmt _NL
2   start: (_NL | stmt)*
3   eval_input: testlist _NL*
4
5   !decorator: "@" dotted_name [ "(" [arguments] ")" ] _NL
6   decorators: decorator+
7   decorated: decorators (classdef | funcdef | async_funcdef)
8
9   async_funcdef: "async" funcdef
10  funcdef: "def" NAME "(" parameters? ")" ["->" test] ":" ( suite | _NL)
11
12  !parameters: paramvalue ("," paramvalue)* ["," [ starparams | kwparams]]
13           | starparams
14           | kwparams
15  starparams: "*" typedparam? ("," paramvalue)* ["," kwparams]
16  kwparams: "**" typedparam
17
18  ?paramvalue: typedparam ["=" test]
19  ?typedparam: NAME [":" test]
20
21  !varargslist: (vfpdef ["=" test] ("," vfpdef ["=" test])* ["," [ "*" [vfpdef] ("," vfpdef
         ["=" test])* ["," ["**" vfpdef [","]]] | "**" vfpdef [","]]]
22    | "*" [vfpdef] ("," vfpdef ["=" test])* ["," ["**" vfpdef [","]]]
23    | "**" vfpdef [","])
24
25  vfpdef: NAME
26
27  ?stmt: (simple_stmt | compound_stmt ) ["eof"]
28  !?simple_stmt: small_stmt (";" small_stmt)* [";"] _NL
29  ?small_stmt: (expr_stmt | del_stmt | pass_stmt | flow_stmt | import_stmt | global_stmt |
         nonlocal_stmt | assert_stmt)
30  ?expr_stmt: testlist_star_expr (annassign | augassign (yield_expr|testlist)
31           | ("=" (yield_expr|testlist_star_expr))*)
32  annassign: ":" test ["=" test]
33  !?testlist_star_expr: (test|star_expr) ("," (test|star_expr))* [","]
34  !augassign: ("+=" | "-=" | "*=" | "@=" | "/=" | "%=" | "&=" | "|=" | "^=" | "<<=" | ">>=" |
         "**=" | "//=")
35  // For normal and annotated assignments, additional restrictions enforced by the interpreter
36  del_stmt: "del" exprlist
37  pass_stmt: "pass"
38  flow_stmt: break_stmt | continue_stmt | return_stmt | raise_stmt | yield_stmt
39  break_stmt: "break"
40  continue_stmt: "continue"
41  return_stmt: "return" [testlist]
42  yield_stmt: yield_expr
43  raise_stmt: "raise" [test ["from" test]]
44  import_stmt: import_name | import_from
45  import_name: "import" dotted_as_names
46  // note below: the ("." | "...") is necessary because "..." is tokenized as ELLIPSIS
47  import_from: "from" (dots? dotted_name | dots) "import" ("*" | "(" import_as_names ")" |
         import_as_names)
48  !dots: "."+
49  import_as_name: NAME ["as" NAME]
50  dotted_as_name: dotted_name ["as" NAME]
51  !import_as_names: import_as_name ("," import_as_name)* [","]
52  dotted_as_names: dotted_as_name ("," dotted_as_name)*
53  dotted_name: NAME ("." NAME)*
54  global_stmt: "global" NAME ("," NAME)*
55  nonlocal_stmt: "nonlocal" NAME ("," NAME)*
56  assert_stmt: "assert" test ["," test]
57
58  compound_stmt: if_stmt | while_stmt | for_stmt | try_stmt | with_stmt | funcdef | classdef |
          decorated | async_stmt
59  async_stmt: "async" (funcdef | with_stmt | for_stmt)
60  if_stmt: "if" test ":" suite ("elif" test ":" suite)* ["else" ":" suite]
61  while_stmt: "while" test ":" suite ["else" ":" suite]
62  for_stmt: "for" exprlist "in" testlist ":" suite ["else" ":" suite]
```

```
63    try_stmt: ("try" ":" suite ((except_clause ":" suite)+ ["else" ":" suite] ["finally" ":"
          suite] | "finally" ":" suite))
64    with_stmt: "with" with_item ("," with_item)*  ":" suite
65    with_item: test ["as" expr]
66    // NB compile.c makes sure that the default except clause is last
67    except_clause: "except" [test ["as" NAME]]
68    suite: simple_stmt | _NL _INDENT stmt+ _DEDENT
69
70    ?test: or_test ["if" or_test "else" test] | lambdef
71    ?test_nocond: or_test | lambdef_nocond
72    lambdef: "lambda" [varargslist] ":" test
73    lambdef_nocond: "lambda" [varargslist] ":" test_nocond
74    ?or_test: and_test ("or" and_test)*
75    ?and_test: not_test ("and" not_test)*
76
77    ?not_test: "not" not_test -> not
78              | comparison
79    ?comparison: expr (_comp_op expr)*
80    star_expr: "*" expr
81    ?expr: xor_expr ("|" xor_expr)*
82    ?xor_expr: and_expr ("^" and_expr)*
83    ?and_expr: shift_expr ("&" shift_expr)*
84    ?shift_expr: arith_expr (_shift_op arith_expr)*
85    ?arith_expr: term (_add_op term)*
86    ?term: factor (_mul_op factor)*
87    ?factor: _factor_op factor | power
88
89    !_factor_op: "+"|"-"|"~"
90    !_add_op: "+"|"-"
91    !_shift_op: "<<"|">>"
92    !_mul_op: "*"|"@"|"/"|"%"|"//"
93    // <> isn't actually a valid comparison operator in Python. It's here for the
94    // sake of a __future__  import described in PEP 401 (which really works :-)
95    !_comp_op: "<"|">"|"=="|">="|"<="|"<>"|"!="|"in"|"not" "in"|"is"|"is" "not"
96
97    ?power: await_expr ["**" factor]
98    !await_expr: ["await"] atom_expr
99
100   ?atom_expr: atom_expr "(" [arguments] ")"      -> funccall
101             | atom_expr "[" subscriptlist "]"  -> getitem
102             | atom_expr "." NAME               -> getattr
103             | atom
104
105   ?atom: "(" [yield_expr|testlist_comp] ")" -> tuple
106        | "[" [testlist_comp] "]"  -> list
107        | "{" [dictorsetmaker] "}" -> dict
108        | NAME -> var
109        | number | string+
110        | "(" test ")"
111        | "..." -> ellipsis
112        | "None"    -> const_none
113        | "True"    -> const_true
114        | "False"   -> const_false
115
116   !?testlist_comp: (test|star_expr) [comp_for | ("," (test|star_expr))+ [","] | ","]
117   !subscriptlist: subscript ("," subscript)* [","]
118   subscript: test | [test] ":" [test] [sliceop]
119   sliceop: ":" [test]
120   !exprlist: (expr|star_expr) ("," (expr|star_expr))* [","]
121   !testlist: test ("," test)* [","]
122   !dictorsetmaker: ( (((test ":" test | "**" expr) (comp_for | ("," (test ":" test | "**" expr)
          )* [","])) | ((test | star_expr) (comp_for | ("," (test | star_expr))* [","])) )
123
124   classdef: "class" NAME ["(" [arguments] ")"] ":" suite
125   !arguments: argvalue ("," argvalue)*  ["," [ starargs | kwargs]]
126            | starargs
127            | kwargs
128            | test comp_for
129
130   !starargs: "*" test ("," "*" test)* ("," argvalue)* ["," kwargs]
131   kwargs: "**" test
132
133   ?argvalue: test ["=" test]
134
135   comp_iter: comp_for | comp_if | async_for
136   async_for: "async" "for" exprlist "in" or_test [comp_iter]
137   comp_for: "for" exprlist "in" or_test [comp_iter]
138   comp_if: "if" test_nocond [comp_iter]
139
140   // not used in grammar, but may appear in "node" passed from Parser to Compiler
141   encoding_decl: NAME
```

```
142
143  yield_expr: "yield" [yield_arg]
144  yield_arg: "from" test | testlist
145
146
147  number: DEC_NUMBER | HEX_NUMBER | OCT_NUMBER | FLOAT_NUMBER
148
149  string: STRING | LONG_STRING
150
151  // Tokens
152  NAME: /[a-zA-Z_]\w*/
153  COMMENT: /#.*(\n[\t ]*)+/ | LONG_STRING
154  _NL: ( /(\r?\n[\t ]*)+/ | COMMENT)+
155
156  LONG_STRING: /[ubf]?r?("""(?<!\\).*?"""|'''(?<!\\).*?''')/is
157
158  DEC_NUMBER: /0|[1-9]\d*/i
159  HEX_NUMBER.2: /0x[\da-f]*/i
160  OCT_NUMBER.2: /0o[0-7]*/i
161  BIN_NUMBER.2 : /0b[0-1]*/i
162  FLOAT_NUMBER.2: /((\d+\.\d*|\.\d+)(e[-+]?\d+)?|\d+(e[-+]?\d+))/i
163
164  %import common.WS_INLINE
165
166  %declare _INDENT _DEDENT
167  %ignore WS_INLINE
168  %ignore /\\[\t \f]*\r?\n/    // LINE_CONT
169  %ignore COMMENT
```

Listing 8: Python Grammar

### A.8.4  Go Grammar

```
1
2   start: package_clause eos (import_decl eos)* ((function_decl | method_decl | declaration)
        eos "eoc"?)*
3
4   package_clause: "package" NAME
5
6   import_decl: "import"  (import_spec | "(" (import_spec eos)* ")")
7
8   import_spec: ("." | NAME)? import_path
9
10  import_path: string_
11
12  declaration: const_decl | type_decl | var_decl
13
14  const_decl: "const"  (const_spec | "(" (const_spec eos)* ")")
15
16  const_spec: identifier_list (type_? "=" expression_list)?
17
18  identifier_list: NAME ("," NAME)*
19
20  expression_list: expression ("," expression)*
21
22  type_decl: "type" (type_spec | "(" (type_spec eos)* ")")
23
24  type_spec: alias_decl | type_def
25
26  alias_decl : NAME "=" type_
27
28  type_def : NAME type_parameters? type_
29
30  type_parameters : "[" type_parameter_decl ("," type_parameter_decl)* "]"
31
32  type_parameter_decl : identifier_list type_element
33
34  type_element : type_term ("|" type_term)*
35
36  type_term : "~"? type_
37
38  // Function declarations
39
40  function_decl: "func" NAME type_parameters? signature ("{" statement_list? ("}" | "eof"))?
41
42  method_decl: "func" receiver NAME signature block?
43
44  receiver: parameters
45
46  var_decl: "var" (var_spec | "(" (var_spec eos)* ")")
47
48  var_spec: identifier_list (type_ ("=" expression_list)? | "=" expression_list)
49
50  block: "{" statement_list? "}"
51
52  statement_list: ((";"? | EOS?) statement eos)+
53
54  statement: declaration | labeled_stmt | simple_stmt | go_stmt | return_stmt | break_stmt |
        continue_stmt | goto_stmt | fallthrough_stmt | block | if_stmt | switch_stmt |
        select_stmt | for_stmt | defer_stmt
55
56  simple_stmt: send_stmt | inc_dec_stmt | assignment | expression | short_var_decl
57
58  send_stmt: expression  "<-" expression
59
60  inc_dec_stmt: expression ("++" | "--")
61
62  assignment: expression assign_op expression | expression_list "=" expression_list
63
64  assign_op: "+=" | "-=" | "|=" | "^=" | "*=" | "/=" | "%=" | "<<=" | ">>=" | "&=" | "&^="
65
66  short_var_decl: expression_list ":=" expression_list
67
68  labeled_stmt: NAME ":" statement?
69
70  return_stmt: "return" expression_list?
71
72  break_stmt: "break" NAME?
73
74  continue_stmt: "continue" NAME?
75
```

```
76   goto_stmt: "goto"   NAME
77
78   fallthrough_stmt: "fallthrough"
79
80   defer_stmt: "defer" expression
81
82   if_stmt: "if"  ( expression | eos expression | simple_stmt eos expression) block ("else" (
         if_stmt | block))?
83
84   switch_stmt: expr_switch_stmt | type_switch_stmt
85
86   expr_switch_stmt: "switch"  (expression? | simple_stmt? eos expression?) "{"
         expr_case_clause* "}"
87
88   expr_case_clause: expr_switch_case ":" statement_list?
89
90   expr_switch_case: "case" expression_list | "default"
91
92   type_switch_stmt: "switch"  ( type_switch_guard | eos type_switch_guard | simple_stmt eos
         type_switch_guard) "{" type_case_clause* "}"
93
94   type_switch_guard: (NAME ":=")? NAME "." "(" "type"  ")"
95
96   type_case_clause: type_switch_case ":" statement_list?
97
98   type_switch_case: "case" type_list | "default"
99
100  type_list: (type_ | "nil" ) ("," (type_ | "nil"  ))*
101
102  select_stmt: "select" "{" comm_clause* "}"
103
104  comm_clause: comm_case ":" statement_list?
105
106  comm_case: "case" (send_stmt | recv_stmt) | "default"
107
108  recv_stmt: (expression_list "=" | identifier_list ":=")? expression
109
110  for_stmt: "for" [for_clause] block
111
112  for_clause: simple_stmt (eos expression eos simple_stmt)? | range_clause
113
114  range_clause: (expression_list "=" | expression_list ":=") "range"  expression
115
116  go_stmt: "go"expression
117
118  type_: literal_type | var_or_type_name type_args? | "(" type_ ")"
119
120  channel_type
121
122  type_args : "--"
123
124  var_or_type_name: NAME "." NAME | NAME | NAME "." "(" type_ ")"
125
126  array_type: "[" array_length "]" element_type
127
128  array_length: expression
129
130  element_type: type_
131
132  pointer_type: "*" type_
133
134  interface_type: "interface" "{" ((method_spec | type_element ) eos)* "}"
135
136  slice_type: "[" "]" element_type
137
138  // It's possible to replace `type` with more restricted type_lit list and also pay attention
         to nil maps
139  map_type: "map" "[" type_ "]" element_type
140
141  channel_type: ("'chan"  | "chan"    "<-" |   "<-" "chan" ) element_type
142
143  method_spec: NAME parameters result | NAME parameters
144
145  function_type: "func" signature
146
147  signature: parameters result?
148
149  result: parameters | type_
150
151  parameters: "(" parameter_decl ("," parameter_decl)* ","? ")" | "(" ")"
152
```

```
153  // a comma-separated list of either (a) name, (b) type, or (c) name and type
154  // parameter_decl: identifier_list? "..."? type_
155
156
157  // Although following is overapproximate it's an easy way to avoid reduce/reduce conflicts
158  parameter_decl: (type_ | "..."? type_ | NAME type_)
159
160
161  expression: primary_expr
162               | ("+" | "-" | "!" | "^" | "*" | "&" | "<-") expression
163               | expression ("*" | "/" | "%" | "<<" | ">>" | "&" | "&^") expression
164               | expression ("+" | "-" | "|" | "^") expression
165               | expression ("==" | "!=" | "<" | "<=" | ">" | ">=") expression
166               | expression "&&" expression
167               | expression "||" expression
168
169  primary_expr: operand | primary_expr ("." (NAME | "(" type_ ")") | index | slice_ |
         arguments) | type_
170
171  // Giving operand higher precedence than type_ is a hack to avoid reduce/reduce conflicts
172  operand.3: literal | NAME | "(" expression ")" // removed NAME type_args?
173
174  literal: basic_lit | composite_lit | function_lit
175
176  basic_lit: "nil" | integer | string_ | FLOAT_LIT | CHAR_LIT
177
178  integer: DECIMAL_LIT | BINARY_LIT | OCTAL_LIT | HEX_LIT
179
180  DECIMAL_LIT: /0|[1-9]\d*/i
181  HEX_LIT.2: /0x[\da-f]*/i
182  OCTAL_LIT.2: /0o[0-7]*/i
183  BINARY_LIT.2 : /0b[0-1]*/i
184  FLOAT_LIT.2: /((\d+\.\d*|\.\d+)(e[-+]?\d+)?|\d+(e[-+]?\d+))/i
185  CHAR_LIT: /'.'/i
186
187  composite_lit: literal_type literal_value
188
189  literal_type: struct_type | array_type | "[" "..." "]" element_type | slice_type | map_type
         | "interface" "{" "}"
190
191  literal_value: "{" (element_list ","?)? "}"
192
193  element_list: keyed_element ("," keyed_element)*
194
195  keyed_element: (key ":")? element
196
197  key: expression | literal_value
198
199  element: expression | literal_value
200
201  struct_type: "struct" "{" (field_decl eos)* "}"
202
203  field_decl: (identifier_list type_ | embedded_field) string_?
204
205  string_: RAW_STRING_LIT | INTERPRETED_STRING_LIT
206
207  RAW_STRING_LIT: /`.*?`/
208  INTERPRETED_STRING_LIT: /".*?"/i
209
210  embedded_field: "*"? (NAME "." NAME | NAME)  type_args?
211
212  function_lit: "func" signature block // function
213
214  index: "[" expression "]"
215
216  slice_: "[" ( expression? ":" expression? | expression? ":" expression ":" expression) "]"
217
218  type_assertion: "." "(" type_ ")"
219
220  arguments: "(" ( expression_list? "..."? ","?)? ")"
221
222  eos: ";" | EOS // | {this.closingBracket()}?
223
224  NAME : /[a-zA-Z_]\w*/
225  EOS: _NL | ";" | "/*' .*? '*/"
226
227  COMMENT : /\/\/[^\n]*/
228  _NL: ( /(\r?\n[\t ]*)+/ | COMMENT)+
229
230  %ignore /[\t ]/
231  %ignore /\\[\t \f]*\r?\n/    // LINE_CONT
```

Listing 9: Go Grammar

