# OpenReview forum: "SynCode: LLM Generation with Grammar Augmentation"
_TMLR — Accepted by TMLR_

### Review · Reviewer_uKiz · 2024-12-30

**Summary Of Contributions:**

The paper contributes a technique for efficiently filtering LLM decoding outputs guided by a context-free grammar and demonstrates this technique substantially increases LLM output in terms of syntactical correctness, while maintaining or increasing the rate of output decoding.

The new technique supports offline precomputation and application to general languages (demonstrated with Python and Go, along with more specialized JSON syntax and SQL). The GPU-based implementation of masking is fast, with evaluation experiments showing the filtered decoding process can be faster than the baseline LLM.

The results show substantial increases to syntactic correctness in JSON output, SQL, python, and Go programming when applied to open models that permit querying of scores. The outputs retain good performance in terms of the correctness of outputs as well.

The new knowledge presented is around the details of the technique as the work is primarily a technical contribution.

**Audience:**

Yes

**Broader Impact Concerns:**

None. Parsers to control decoding could theoretically be used to guide LLMs to produce socially harmful outputs, but this capability is not particularly new to the space of using LLMs or unique to the SynCode implementation.

**Claims And Evidence:**

Yes

**Requested Changes:**

# critical
- Add reports of statistical variability in the results over multiple runs.
	- The key speedup results are very large and likely not subject to noise, but additional runs and variability reports would strengthen many of the other reported outcomes (like speed and correctness).

# strengthen
- Add functional testing results to SQL.
- Clarify the SQL testing design.
	- Table 2 reports the percentage of syntactically correct queries by difficulty type. But it only reports a single value for the runtime execution success. Should this also be divided by difficulty? Or did I misunderstand the setup?
	- "For example, with the Llama-3.2-3B model, SynCode achieves an execution success rate of 81.4%, compared to 67.4% for unconstrained generation"
		- Is this success rate among syntactically valid samples, or all samples?
- Section 6.1 reports on llama.cpp but uses the CPU, unlike the GPU used for all other results. This is misleading and should be omitted, rather than including the unfair comparison. Ideally the experiments would include llama.cpp using a GPU, but if not, it would be fairer to simply not include the result.
- Section 6.4: "We see smaller generation time and memory with Llama-2-7B-chat and JSON grammar as opposed to LLaMA-7B, WizardCoder-1B, and CodeGen-350M with Python and Go grammars since the size of the mask store is proportional to the number of terminals in the grammar."
	- Where are these results reported? I think I missed them.
- Is SynCode going to be released? I did not see information on that.

**Strengths And Weaknesses:**

# strengths
- Demonstrates large margins of improvement on syntactic correctness in very practically relevant problem settings for LLM generation (Python, Go, SQL, and JSON output).
- Shows no major trade-offs in runtime performance and only modest costs in offline compute and memory needs.


# weaknesses
- The paper includes no statistical results on variability.
	- Many differences are small, specifically for runtime speed and functional correctness in Python and Go.
- The method cannot be applied to languages that do not expose scores.
- (very minor) Readers need modest background in programming languages and parsing to follow the text.

---

> ### Author Response · Authors · 2025-02-06
>
> > Add reports of statistical variability in the results over multiple runs.
> The key speedup results are very large and likely not subject to noise, but additional runs and variability reports would strengthen many of the other reported outcomes (like speed and correctness).
>
> Thank you for this suggestion. We have added standard errors for runtime comparison in Table 1 and correctness results in Table 4. As pointed out by the reviewer the speedup results are large and not subject to substantial noise. Additionally, since the evaluations in Table 1 (JSON) and Table 2 (SQL) use deterministic greedy decoding, these results are identical across runs. Therefore, error bars are unnecessary for the accuracy numbers reported in these tables.
>
> > Add functional testing results to SQL.
> Clarify the SQL testing design. Table 2 reports the percentage of syntactically correct queries by difficulty type. But it only reports a single value for the runtime execution success. Should this also be divided by difficulty? Or did I misunderstand the setup?
>
> There is a misunderstanding here. The table reports execution accuracy measured using the tests in the Spider dataset and not the syntactic accuracy. We have made this more explicit in the update of the paper in Section 6.2.
>
>
> > "For example, with the Llama-3.2-3B model, SynCode achieves an execution success rate of 81.4%, compared to 67.4% for unconstrained generation."
> Is this success rate among syntactically valid samples, or all samples?
>
> The execution success rate is among all the samples.
>
>
> > Section 6.1 reports on llama.cpp but uses the CPU, unlike the GPU used for all other results.
> This is misleading and should be omitted, rather than including the unfair comparison. Ideally, the experiments would include llama.cpp using a GPU, but if not, it would be fairer to simply not include the result.
>
> We have removed llama.cpp results from our evaluation.
>
>
> > Section 6.4: "We see smaller generation time and memory with Llama-2-7B-chat and JSON grammar as opposed to LLaMA-7B, WizardCoder-1B, and CodeGen-350M with Python and Go grammars since the size of the mask store is proportional to the number of terminals in the grammar."
> Where are these results reported? I think I missed them.
>
> Mask store generation time for Python and Go is in Table 5 and for JSON it is discussed in Section 6.4. The number of terminals is mentioned in Section 5. In the updated version, we have added the number of terminals in Section 6.4 again to avoid any confusion.
>
> > Is SynCode going to be released? I did not see information on that.
>
> Yes, the SynCode code is already available publicly. We excluded the link in the submitted version of the paper for anonymity. We will include the link in the final version of the paper.

---

> > ### Comment · Reviewer_uKiz · 2025-03-02
> >
> > Thank you for addressing my comments, the paper looks much improved!
> >
> > I have no further remarks or requests.

---

### Review · Reviewer_cP3J · 2025-01-11

**Summary Of Contributions:**

The authors introduce a method for constraining LLMs with grammars by precalculating DFA mask store for any language with CFG. They showcase the method on different languages such as JSON, Python, and GO. The authors also provide proof of completeness of their method.

**Audience:**

Yes

**Claims And Evidence:**

Yes

**Requested Changes:**

See weaknesses.

**Strengths And Weaknesses:**

Strengths

I appreciate the effort the authors took to explain the method with examples. Besides, the overall paper is well-written, and should be a good read for a more general audience. The experiments and proofs validate the authors claim too.

Weakness

My only concern is that how the pre-calculation step scales. While the authors claim that the size of terminals is reasonable in practice, it will be interesting to see how the time complexity scales with larger $|\mathcal{A}|$. Another question I feel the authors should answer is how the space complexity is (and additionally, how they scale).

---

> ### Author Response · Authors · 2025-02-06
>
> > My only concern is how the pre-calculation step scales. While the authors claim that the size of terminals is reasonable in practice, it will be interesting to see how the time complexity scales with larger $|\mathcal{A}|$.
>
> We evaluate reasonably large grammars in our experiments, such as Python (94 terminals) and Go (87 terminals), and observe that constructing the DFA mask store consistently takes less than 10 minutes with accept sequences of size 1 and 2. As the reviewer pointed out, with larger accept sequences, the pre-calculation step may take more time. However, this is a one-time cost per model and grammar. All subsequent uses of the mask store incur no additional cost. Additionally, for all evaluated tasks we found sequences of size less than 2 sufficient for eliminating syntax errors.
>
> Our current implementation is Python-based, and there is scope to optimize this construction of the mask store with better parallelization in faster languages. Further, a good portion of possible terminal sequences are potentially invalid according to the grammar. Thus, while considering a larger lookahead of terminals, we could selectively store mapping for terminal  sequences that are acceptable according to the grammar in the mask store. We defer these optimizations useful for larger accept sequences to the future work.
>
>
>
> > Another question I feel the authors should answer is how the space complexity is.
>
> Thank you for the suggestion. We have added space complexity discussion in Section 4.6.

---

### Review · Reviewer_G81y · 2025-04-03

**Summary Of Contributions:**

This paper introduces SynCode, a framework that enhances the accuracy when making LLMs generate some specific kinds of code language. SynCode augment the LLMs with an additional (pre-computed) look-up table, named Deterministic Finite Automation (DFA) mask store to impose restriction on the LLM generation process. The DFA mask store maps states of the DFA to boolean masks over the LLM's vocabulary. These masks indicate which tokens are syntactically valid at each step of the generation process.

**Audience:**

Yes

**Claims And Evidence:**

Yes

**Requested Changes:**

1. I would suggest the authors to highlight the key contribution of SynCode and its key difference with previous works (e.g., DOMINO and Outlines), in terms of performance, computation and storage overhead. Empirical evidences are highly preferable.

2. The reviewer suggests the authors to add more case study to show what kind of errors that SynCode may occur. What about in comparison with DOMINO, Outlines and other similar approaches based on constraint decoding and code language grammar induction?

**Strengths And Weaknesses:**

**Strengths:**

(1): SynCode ensures that all valid tokens are retained and all invalid tokens are filtered out at each generation step, guaranteeing syntactic correctness.

(2): The framework minimizes runtime overhead by leveraging precomputed masks and efficient hardware utilization, such as GPU acceleration.

(3): SynCode can be integrated with any language defined by a CFG and works seamlessly with various LLM decoding strategies, including greedy decoding, beam search, and different sampling methods.

(4): The paper is overall well-writen and easy-to-follow.

**Weaknesses:**

(1): I think the most important weaknesses for the SynCode is that the underlying idea of the framework (i.e., impose restriction on the LLM generation process through masking out all of the irrelevant tokens) is widely used in the previous works (e.g., Beurer-Kellner et al., 2024 DOMINO, Willard and Louf, 2023 Outlines, ...). Note that the authors have discussed the difference in the Related Work section and Table 6. From my point of view, those ideas may have very minor difference. Hence I would suggest the authors to highlight the key technical difference and the contribution.

(2): Generating a DFA mask may require additional processing time and memory.

---

> ### Author Response · Authors · 2025-04-06
>
> > I would suggest the authors to highlight the key contribution of SynCode and its key difference with previous works (e.g., DOMINO and Outlines).
>
> While SynCode shares the primary goal of constraining LLM generation through token masking with prior works like DOMINO and Outlines, there are several significant technical differences that constitute novel contributions:
>
> **Decoding Efficiency**: One of the key novelties of SynCode's is in how it processes acceptable terminals during decoding - by handling each sequence independently rather than computing expensive unions of FSMs (as in Outlines) or traversing prefix trees (as in DOMINO). This architectural difference yields substantial efficiency gains during inference.
>
> **DFA Mask Store caching**: Our DFA mask store data structure leverages Boolean masks that can be combined efficiently on modern hardware. This structure is unique to SynCode and makes it more efficient than Domino’s prefix tree structure.
>
> **Over-approximation vs. Under-approximation**: SynCode's conservative over-approximation approach (allowing tokens whose prefixes match the parser lookahead) provides soundness guarantees with any lookahead depth. This contrasts with DOMINO's under-approximation approach, which requires infinite parser lookahead for equivalent guarantees.
>
> Additionally, our work is first to rigorously demonstrate the soundness and completeness of syntactical LLM generation. While Domino claims soundness under infinite parser lookahead, it does not provide a mathematical proof, unlike our approach.
>
> We have further emphasized on these points in our Related Work discussion.
>
> >  Empirical evidences are highly preferable.
> As shown in our evaluation Section 6.1, SynCode outperforms Outlines and many other grammar-constrained LLM generation baselines. Domino’s code is not available publicly to experimentally compare it with SynCode.
>
> We have added comparison to Outlines (latest version https://github.com/dottxt-ai/outlines) in SQL generation on the Spider datase on the first 100 examples. We see that SynCode outperforms Outlines significantly across all models tested by improved accuracy and 10x faster inference time.
>
> | Model | Outlines |  | Syncode |  |
> |-------|--------------|------------|-------------|-------------|
> |  | **Accuracy** | **Avg Time (s)** | **Accuracy** | **Avg Time (s)** |
> | Llama-3.2-1B | 4% | 7.88 | 25% | 0.70 |
> | Llama-3.2-3B | 17% | 11.16 | 31% | 1.03 |
> | Gemma-2-2b-it | 21% | 45.82 | 25% | 5.81 |
>
>
> > Generating a DFA mask may require additional processing time and memory.
>
> Our algorithm incurs a preprocessing cost one time for each model and then the mask store can be reused for all subsequent inferences. There are potential optimizations that could reduce both size and computation of the mask store, including filtering out only grammar-valid terminal sequences and implementing the solution in more efficient languages like Rust. We have deferred these optimizations for future work.
>
> > The reviewer suggests the authors to add more case study to show what kind of errors that SynCode may occur. What about in comparison with DOMINO, Outlines and other similar approaches based on constraint decoding and code language grammar induction?
>
> In Appendix A.6, we present the common errors observed with SynCode when generating Python code. These errors typically occur when the model maintains grammatically correct prefix (theoretically sound) but fails to terminate properly. Since Domino's code is unavailable and Outlines was incompatible with our large Python/Go grammars, we were unable to make direct comparisons with these alternative approaches on code generation.

---

### Decision · Action_Editor_7S6n · 2025-05-03

**Recommendation:** Accept with minor revision

**Comment:**

All three reviewers provided positive evaluations, recommending accept or leaning accept, with suggestions largely focused on empirical completeness and clarity of comparison:
- Reviewer G81y emphasized the novelty in proving both soundness and completeness, but requested clearer differentiation from DOMINO and Outlines.
- Reviewer cP3J asked for clarification on scalability and complexity of the DFA mask store, which the authors addressed by reporting time/memory requirements for Python and Go grammars.
- Reviewer uKiz appreciated the practical value but suggested adding statistical variability, clarifying SQL execution metrics, and removing unfair comparisons with CPU-based baselines.

The authors have adequately revised the paper to address all major concerns. The authors are recommended to include the clarifications addressed during the response period into the camera-ready version.

**Audience:**

The paper is well-targeted for the TMLR audience, especially those interested in: (1) language model decoding algorithms, (2) program synthesis and code generation, (3) formal language theory applications in ML, and (4) efficient inference in large models.

**Claims And Evidence:**

The submission presents a technically sound and empirically validated framework—SynCode—for improving syntactic correctness in LLM generation via grammar-constrained decoding. The authors rigorously prove soundness and completeness of their decoding method with respect to a context-free grammar and support their claims through extensive empirical evaluation across diverse generation tasks including JSON, SQL, Python, and Go. Notably, the approach introduces a novel DFA Mask Store, which precomputes token validity and enables efficient decoding through masking. The method is benchmarked against baselines such as Outlines, with results demonstrating (1) Elimination of syntax errors in JSON generation. (2) 96.07% reduction in syntax errors for Python and Go. (3) 10× decoding speedup over Outlines in SQL generation. The authors also address scalability concerns of DFA preprocessing and memory, and provide empirical results confirming efficiency even on grammars with 90+ terminals. While some prior work (e.g., DOMINO, Outlines) shares conceptual similarities in constrained decoding, SynCode is distinguished by its formal guarantees and computational efficiency.